# Does CLIP's generalization performance mainly stem from high train-test similarity?

**Prasanna Mayilvahanan**[1,2,3*]   **Thaddäus Wiedemer**[1,2,3*]   **Evgenia Rusak**[1,2,3]

**Matthias Bethge**[1,2]   **Wieland Brendel**[2,3,4]

[1]University of Tübingen    [2]Tübingen AI Center
[3]Max-Planck-Institute for Intelligent Systems, Tübingen    [4]ELLIS Institute Tübingen

prasanna.mayilvahanan@uni-tuebingen.de, thaddaeus.wiedemer@gmail.com

## Abstract

Foundation models like CLIP are trained on hundreds of millions of samples and effortlessly generalize to new tasks and inputs. Out of the box, CLIP shows stellar zero-shot and few-shot capabilities on a wide range of out-of-distribution (OOD) benchmarks, which prior works attribute mainly to today's large and comprehensive training dataset (like LAION). However, it is questionable how meaningful CLIP's high zero-shot performance is as it seems likely that web-scale datasets like LAION simply contain many samples that are similar to common OOD benchmarks originally designed for ImageNet. To test this hypothesis, we retrain CLIP on pruned LAION splits that replicate ImageNet's train-test similarity with respect to common OOD benchmarks. While we observe a performance drop on some benchmarks, surprisingly, CLIP's overall performance remains high. This shows that high train-test similarity is insufficient to explain CLIP's performance, and other properties of the training data must drive CLIP to learn good representations. Additionally, by pruning data points that are dissimilar to the OOD benchmarks, we uncover a 100M split of LAION (¼ of its original size) on which CLIP can be trained to match its original performance.

## 1 Introduction

Large models like GPT-4 (OpenAI, 2023; Schulman et al., 2022), CLIP (Radford et al., 2021), or LLaMa (Touvron et al., 2023) are changing the technological and academic landscape with their unprecedented performance and breadth of viable applications. A core characteristic of these *Foundation Models* (Bommasani et al., 2021) is that they are trained on hundreds of millions or even billions of data points scraped from the internet. For example, OpenCLIP (Schuhmann et al., 2022), the open-source version of CLIP (Radford et al., 2021), is trained on LAION-400M, a web-scale dataset with a wide variety of image-text pairs (Schuhmann et al., 2021). CLIP forms the backbone of generative models like DALL-E2 (Ramesh et al., 2022) and is known for its remarkable zero-shot and few-shot performance on a wide range of tasks, specifically on out-of-distribution (OOD) benchmarks like ImageNet-Sketch (Wang et al., 2019), ImageNet-R (Hendrycks et al., 2020), etc.

Prior work has shown that CLIP's stellar performance stems mainly from its data distribution (Fang et al., 2022; Radford et al., 2021). Nevertheless, it remains unclear which specific properties of the training distribution, such as its scale, diversity, density, or relation to the test set, drive performance. OOD benchmarks like ImageNet-Sketch and ImageNet-R were initially designed in reference to ImageNet-1k (Deng et al., 2009), which had served as the primary dataset driving progress in machine vision for several years before the emergence of web-scale datasets. ImageNet-Sketch, ImageNet-R, and others are considered OOD because they share the same content (i.e., classes) as ImageNet-1k but are *dissimilar* in terms of style, pose, scale, background, or viewpoint. There is no guarantee that

---

*Equal contribution. Code available at https://github.com/brendel-group/clip-ood

these datasets are also *dissimilar* to LAION-400M. We provide evidence in Fig. 1 where we choose samples from ImageNet-Sketch and ImageNet-R and examine their nearest perceptual neighbors in LAION-400M and ImageNet-Train. We find highly *similar* neighbors and even exact duplicates in LAION-400M while neighbors in ImageNet-Train are relatively *dissimilar*. In other words, models trained on LAION-400M may perform well on conventional OOD benchmarks simply due to being trained on semantically and stylistically *similar* data points. Naturally, the question arises:

*Does CLIP's accuracy on OOD benchmarks mainly stem from highly similar images in its train set?*

By *highly similar images*, we mean images that are stylistically and semantically more similar to the test sets than any image in ImageNet-1k is. To answer this question, we make the following contributions:

- In Sec. 4.1, we begin by introducing *perceptual similarity* (Ilharco et al., 2021), which has previously been shown to capture stylistic and semantic similarity between images (Fu et al., 2023; Gadre et al., 2023; Zhang et al., 2021). We show in Sec. 4.2 that the similarity of nearest neighbors under this metric generally impacts CLIP's performance. Specifically, we (i) observe a high correlation between zero-shot accuracy and nearest-neighbor similarity of test samples and (ii) demonstrate that similarity-based pruning of the training set greatly affects CLIP's performance.

- Based on these insights, we compare the distribution of nearest-neighbor similarities of different training sets in Sec. 4.3 and find that they differ substantially. We hypothesize that CLIP's high performance might be largely explained by the training samples that cause this difference, which we term *highly similar images*.

- Sec. 4.4 formalizes the notion of *highly similar images* based on the *similarity gap* of two training distributions. Under this formalization, *highly similar images* of LAION-400M lie within the similarity gap of ImageNet-Train to a given test set, i.e., are more similar to test samples than any image in ImageNet-Train is. We go on to show how pruning can align the similarity gap of both distributions, such that test sets are as dissimilar to pruned LAION-400M-splits as they are to ImageNet-Train.

- As our central result in Sec. 5, we surprisingly find that training CLIP on the curated subsets only marginally decreases performance on the corresponding OOD benchmarks (Tab. 1). We conclude that high train-test similarity cannot fully explain CLIP's remarkable performance, and other properties of LAION-400M must play a role.

- To facilitate future research into the impact of training on the performance of vision-language foundation models, we curate a 100M subset of LAION-400M (¼ of its original size) on which CLIP maintains its full OOD benchmark performance (Sec. 4.2 & B.4).

## 2 RELATED WORK

**Measuring OOD generalization**  To assess expected model performance in the wild, researchers use different test sets that are considered OOD with respect to the training distribution. The terms OOD generalization, (distributional) robustness, or just generalization are used interchangeably by the community. This work mainly focuses on standard datasets that share classes with ImageNet. They include: image renditions (ImageNet-R; Hendrycks et al., 2020), unusual camera views and object positions (ObjectNet; Barbu et al., 2019), images selected to be difficult for ImageNet-trained ResNet-50s (ImageNet-A; Hendrycks et al., 2021) and sketches of ImageNet classes (ImageNet-Sketch; Wang et al., 2019). We also consider two datasets commonly considered in-distribution, namely ImageNet-Val (Deng et al., 2009), and ImageNet-V2 (Recht et al., 2019).

**ID vs. OOD generalization**  While researchers treat the test sets listed above as OOD with respect to the training distribution when they study robustness, this core assumption is rarely scrutinized. Large-scale language-image models such as CLIP (Radford et al., 2021), ALIGN (Jia et al., 2021), or BASIC (Pham et al., 2021) claim exceptional OOD generalization and zero-shot capabilities. Fang et al. (2022) probe which aspects of the models—like language supervision, cost function, or training distribution—are related to a model's effective OOD robustness and find that differences in the distribution play a key role. Further, Nguyen et al. (2022) find that combining data from multiple sources for training interpolates the model's effective robustness on an OOD test set between the

performance of the model trained on either data source. Here, we aim to extend the findings of Fang et al. (2022) and Nguyen et al. (2022) by evaluating whether high similarity between training and test set is the main driver of CLIP's claimed performance, or whether CLIP is truly better at generalizing across larger distribution shifts.

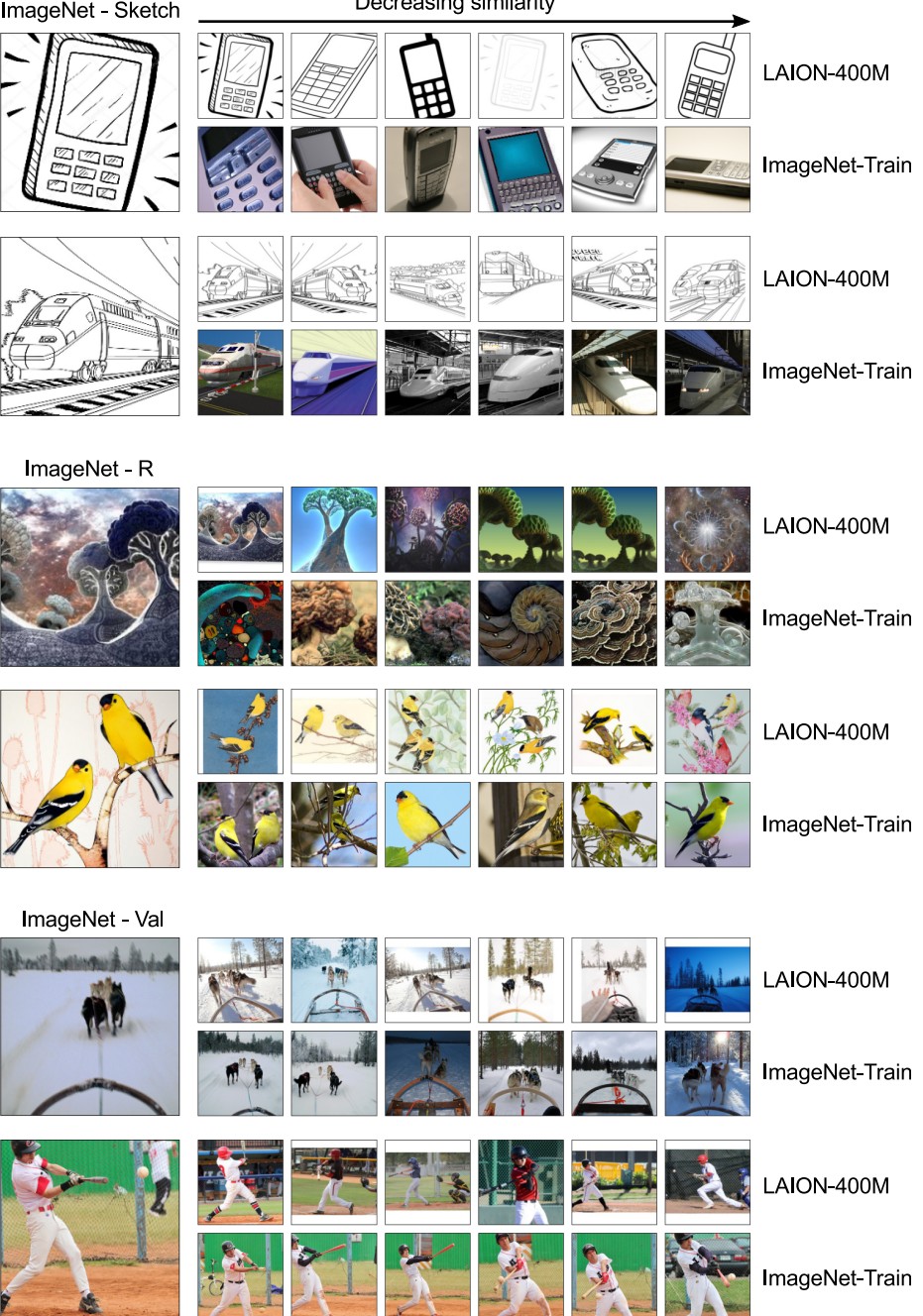

Figure 1: **Similarity of common benchmarks to LAION-400M and ImageNet-Train.** We show nearest neighbors of ImageNet-Sketch, ImageNet-R and ImageNet-Val samples in LAION-400M and ImageNet-Train ordered by decreasing *perceptual similarity*. We omit duplicates within these nearest neighbors. Perceptual similarity is cosine similarity computed in CLIP's image embedding space (see Sec. 4) and can be thought of as measuring the perceptual closeness of images in terms of content and style. LAION-400M clearly contains more similar images to samples from ImageNet-Sketch and ImageNet-R, in contrast ImageNet-Train is more similar to ImageNet-Val. More details in App. G.

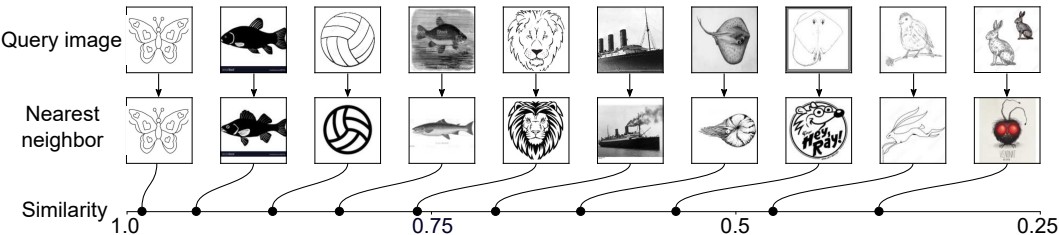

Figure 2: **Relation between *perceptual similarity* and visual closeness of nearest neighbors**. Query images are sampled from ImageNet-Sketch (top row) and are connected to their nearest neighbor in LAION-400M (bottom row). As in Fig. 1, perceptual similarity is simply the cosine similarity measured in CLIP ViT-B/16+'s image embedding space.

## 3 EXPERIMENTAL DETAILS

This section contains technical specifics of image-to-image similarity computation, training details, deduplication, and LAION-200M. Readers can skip this section and return to it when they seek details on the aforementioned. For computing image-to-image similarity, measuring duplicates, and pruning data points, we use CLIP ViT-B/16+'s image embedding space. For all our pruning experiments, we train CLIP ViT-B/32 (Dosovitskiy et al., 2020) for 32 epochs with a batch size of 33,600 on one node with eight A100 GPUs (training takes several days, depending on the dataset size). We use the implementation provided by Ilharco et al. (2021) and stick to their settings for learning rate, weight decay, etc. Our downloaded version of LAION-400M contains only 377M images overall due to missing or broken links, compared to the original 400M used in OpenCLIP (Ilharco et al., 2021).

**LAION-200M**  Abbas et al. (2023) show that pruning exact duplicates, near duplicates, and semantically very similar samples *within* LAION-400M (not yet taking any test sets into account) can reduce dataset size by up to $50\%$ without performance degradation. We re-implement their method to generate our baseline LAION split containing 199M samples, which we refer to as LAION-200M. This step is important to make training multiple instances of CLIP feasible, and we observe that the incurred drop in performance is negligible (compare Tab. 1).

## 4 THE SIMILARITY HYPOTHESIS

This section first illustrates how perceptual similarity can be quantified (Sec. 4.1). Based on this metric, we demonstrate that CLIP's performance on a test set is strongly related to the *nearest-neighbor similarity* between LAION-400M and a test set (Sec. 4.2). Further, we show that nearest-neighbor similarities differ between LAION-400M and ImageNet-Train, which leads to the hypothesis that this difference explains CLIP's high classification accuracy on ImageNet-based test sets (Sec. 4.3). Finally, we phrase this hypothesis in terms of *highly similar images*, which leaves us with an interventional method to test this hypothesis (Sec. 4.4).

### 4.1 QUANTIFYING PERCEPTUAL SIMILARITY

Abbas et al. (2023) demonstrated that nearest neighbors in the image embedding space of CLIP share *semantic* and *stylistic* characteristics. We illustrate this in Fig. 2, where we plot samples from ImageNet-Sketch and their nearest neighbors in LAION-400M for different similarity values. Visually, the similarity scores correlate well with the closeness of the image pairs. This is corroborated by other works that demonstrate high perceptual alignment between CLIP's embedding similarity and human perception (Fu et al., 2023), or using it to sample ImageNet-like images from a large dataset (Gadre et al., 2023), or building a similarity-based classifier (Zhang et al., 2021).

We follow these works and quantify *perceptual similarity* as the cosine similarity in CLIP ViT-B/16+'s image embedding space. App. E ablates the choice of the model used to compute this metric. We denote the similarity of two samples $x_i, x_j \in \mathbb{R}^n$ as

$$s(x_i, x_j) : \mathbb{R}^n \times \mathbb{R}^n \to [-1, 1]. \tag{1}$$

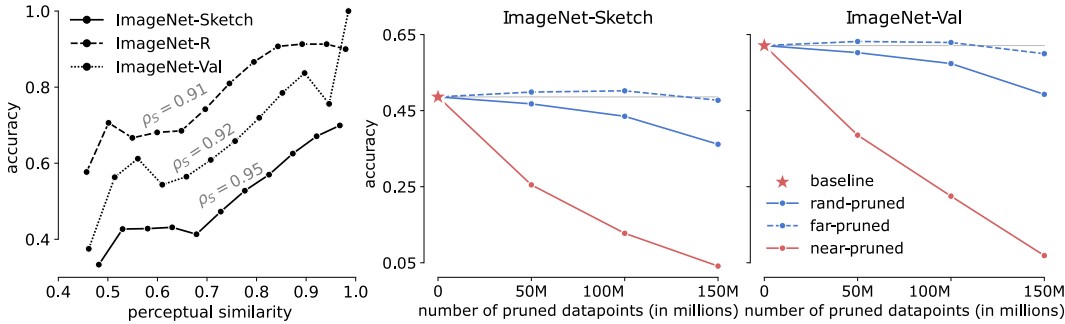

Figure 3: **Nearest-neighbor similarity is predictive of performance**. **Left**: LAION-400M-trained CLIP's top-1 classification accuracy on test samples is highly correlated to their nearest-neighbor similarity $s_{\text{test},i}$. Results are averaged over 0.05 similarity intervals. **Center and right**: Similarity-based pruning greatly impacts CLIP's top-1 classification accuracy. We train a baseline model on LAION-200M (see Sec. 3) and additional models on LAION-200M-splits created by random pruning, near-pruning (in order of decreasing similarity), and far-pruning (in order of increasing similarity). Compared to training on 'rand-pruned' splits (solid blue curve), training on 'near-pruned' splits (solid red curve) drastically decreases classification accuracy. Training on 'far-pruned' splits (dashed blue curve) impacts accuracy comparatively little.

We now consider the relation between a training dataset $\mathcal{D}$ and a test set $\mathcal{T}$. Using the similarity metric $s$, we can find the nearest neighbor in the test set for each training sample. This allows us to assign each training sample $x_i \in \mathcal{D}$ the *nearest-neighbor similarity*

$$s_{\text{train},i}(\mathcal{D}, \mathcal{T}) = \max_{t \in \mathcal{T}} s(t, x_i). \tag{2}$$

In the same way, we can assign each test sample $t_i \in \mathcal{T}$ the *nearest-neighbor similarity*

$$s_{\text{test},i}(\mathcal{D}, \mathcal{T}) = \max_{x \in \mathcal{D}} s(t_i, x). \tag{3}$$

### 4.2 NEAREST-NEIGHBOR SIMILARITY DRIVES PERFORMANCE

We can now examine the relationship between nearest-neighbor similarity and CLIP's zero-shot classification performance.

Fig. 3 (left) illustrates that the nearest-neighbor similarity $s_{\text{test},i}$ of test samples in ImageNet-Sketch, ImageNet-R, and ImageNet-Val to LAION-200M is a good predictor of CLIP's top-1 accuracy on these samples. We observe a clear correlation between nearest-neighbor similarity and accuracy across datasets. For ImageNet-Sketch, for example, sketches without similar counterparts in LAION-400M (similarity 0.38) are classified with $35\%$ accuracy, while sketches duplicated in LAION-400M (similarity close to 1) reach up to $69\%$ accuracy. We show additional correlation plots for ImageNet-based test sets in App. B and for other test sets in App. D.

We can observe the impact of nearest-neighbor similarity on classification performance more directly by pruning samples from LAION-200M based on their nearest-neighbor similarity $s_{\text{train},i}$ to a given test set, retraining CLIP, and evaluating its zero-shot classification performance on that test set. We compare three different pruning strategies: 'near-pruning' prunes in decreasing order of similarity (pruning samples with high nearest-neighbor similarity first), 'far-pruning' prunes in increasing order of similarity, and 'rand-pruning' prunes randomly irrespective of similarity. All strategies produce LAION-200M-splits with 50M, 100M, and 150M pruned samples.

CLIP's zero-shot classification performance when trained on these splits is illustrated in Fig. 3 for ImageNet-Sketch and ImageNet-Val. The 'near-pruned' accuracy curve drops much quicker with decreasing dataset size than the 'rand-pruned' curve. This reiterates that CLIP's classification performance is directly related to the similarity of its training set to the test set. Additional visualizations for other datasets (both ImageNet-based and otherwise) as well as a comparison with ImageNet-trained models can be found in Apps. B and D. Note that since we prune large fractions of the training set here, the pruned images are not yet very specific to the test set used to compute $s_{\text{train},i}$. As a

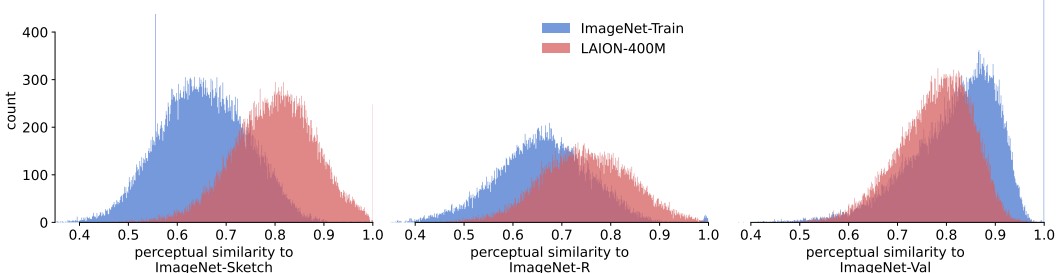

Figure 4: **Nearest-neighbor similarity distribution differs between LAION-400M and ImageNet-Train**. The histograms display the similarity $s_{\text{test},i}$ of samples in ImageNet-Sketch (left), ImageNet-R (center), and ImageNet-Val (right) to their nearest neighbors in LAION-400M (red) and ImageNet-Train (blue). ImageNet-Sketch and ImageNet-R are overall more similar to LAION-400M, while ImageNet-Train is more similar to ImageNet-Val.

result, pruning based on one ImageNet-based dataset generally decreases performance across many ImageNet-based datasets, although not on not on other tasks (see App. B).

The observation so far is not surprising: Performance on the test set decreases in tandem with the training distribution's similarity to the test set. However, our results validate using similarity-based pruning as an effective intervention that allows us to study how training samples impact performance on a given test set. In the next sections, we will explore how to hone this method to arrive at a more precise conclusion about the role of *highly similar images*.

**Core set**    As an aside, we notice that CLIP's performance when trained on 'far-pruned' LAION-200M-splits remains stable up until a dataset size of 100M (see Fig. 3). The performance even slightly surpasses the baseline, further indicating that dissimilar samples do not contribute to CLIP's performance and instead act more like noise in the training data. Motivated by this performance, we extract a LAION-400M *core set* with only 100M images by 'far-pruning' based on not one but six common ImageNet-based benchmarks simultaneously. CLIP trained on this core set outperforms models trained on a de-duplicated dataset of the same size (Ilharco et al., 2021) and roughly matches the performance of a LAION-200M-trained model (see Appx. B.4). We release this core set to ease further exploration of the relationship between training distribution and CLIP's zero-shot performance.

### 4.3    COMPARING NEAREST-NEIGHBOR SIMILARITIES BETWEEN TRAINING SETS

Given the impact of nearest-neighbor similarity on CLIP's zero-shot performance, it is natural to ask how LAION-400M's nearest-neighbor similarity compares to that of other datasets. Specifically, for ImageNet-based benchmarks like ImageNet-Sketch and ImageNet-R, we compare the distribution of nearest-neighbor similarities $s_{\text{test},i}$ to LAION-400M and ImageNet-Train. We have already seen in Fig. 1 that compared to ImageNet-Train, LAION-400M seemed stylistically and semantically much more similar to ImageNet-Sketch and ImageNet-R, while the effect was reversed for ImageNet-Val. Using the notion of perceptual nearest-neighbor similarity, we can now fully capture the difference in similarity in a principled manner. This is illustrated in Fig. 4, where we can now clearly observe that compared to ImageNet-Train, LAION-400M is indeed overall more similar to ImageNet-Sketch and ImageNet-R. We show additional histograms for other test sets in Apps. B and D.

Moreover, in Appx. A.2, we detail how many training samples in LAION-400M and ImageNet-Train are *near duplicates* (duplicates up to small shifts or crops) of the test sets. While we found $3.1\%$ of ImageNet-Sketch images to have duplicates in LAION-400M, there are only $0.04\%$ ImageNet-Sketch duplicates in ImageNet-Train. On the other hand, ImageNet-Train contains duplicates of $2.67\%$ ImageNet-Val images as opposed to just $0.14\%$ ImageNet-Val images in LAION-400M.

LAION-400M-trained CLIP has been reported to outperform ImageNet-trained methods on ImageNet-Sketch and ImageNet-R, while underperforming on ImageNet-Val (see Tab. 1). In light of the above observation, this could well be explained not by LAION-400M's general scale and diversity but specifically by its fraction of training samples whose nearest-neighbor similarity to the test set surpasses that of *any* sample in ImageNet-Train. We term those samples *highly similar images*. The

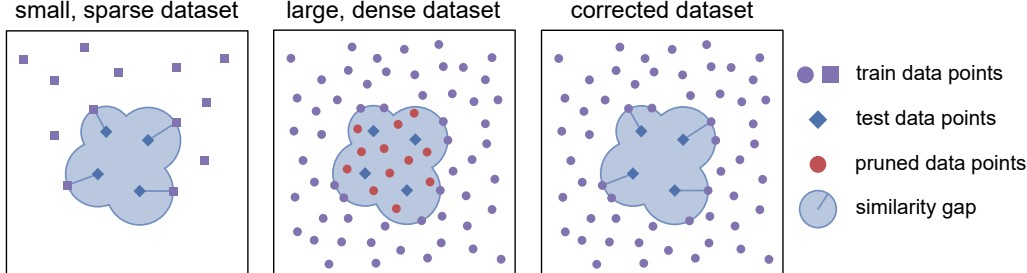

Figure 5: **Aligning the similarity gap of two datasets**. A larger, denser, more diverse dataset likely contains samples more similar to given test points than a smaller, sparser one. To control for this, we compute the nearest-neighbor similarity of each test point to the smaller dataset (left) and prune points from the larger dataset that lie within this hull (center). We end up with a corrected large dataset replicating the *similarity gap* of the small one (right).

following section formalizes this concept and explains how we can refine the similarity-based pruning from Sec. 4.2 to quantify their impact on CLIP's zero-shot classification performance.

### 4.4 SIMILARITY GAP AND HIGHLY SIMILAR IMAGES

Secs. 4.2 and 4.3 provide direct and indirect evidence that CLIP's performance on common ImageNet-based benchmarks might mainly stem from images in its training set that are *highly similar* to the test sets. We now formalize this notion and describe how to systematically test our hypothesis. To this end, we note that even for ImageNet-Train, the nearest-neighbor similarity $s_{\text{test},i}$ differs across test samples. Our goal is to prune LAION-400M so that the pruned dataset replicates the nearest-neighbor similarities $s_{\text{test},i}$ of ImageNet-Train.

Let us consider that we now have two training datasets, denoted as $\mathcal{D}_S$ (small, like ImageNet-Train) and $\mathcal{D}_L$ (large, like LAION-400M), and still use a test dataset $\mathcal{T}$ (like ImageNet-Sketch). For the sake of simplicity, we assume that $\mathcal{D}_S$ is a subset of $\mathcal{D}_L$. We choose a similarity measure $s$ as in Sec. 4.2. We collect all nearest-neighbor similarities $s_{\text{test},i}$ (recall Eq. 3) in the set

$$S(\mathcal{D}, \mathcal{T}) = \big\{ s_{\text{test},i}(\mathcal{D}, \mathcal{T}) \mid i \in \big[1, |\mathcal{T}|\big] \big\} \tag{4}$$

which we term *similarity gap*. We can think of this set as a full characterization of the training set's similarity to any point in the test set; compare Fig. 5.

Based on the assumption that the large dataset contains all samples from the small dataset, it follows that $s_i(\mathcal{D}_S) \leq s_i(\mathcal{D}_L)$. In other words, the nearest-neighbor similarity to samples in the small training set is always smaller than or equal to the similarity to samples in the large training set. Consequently, on a per-sample basis, $S(\mathcal{D}_L, \mathcal{T})$ is strictly larger than $S(\mathcal{D}_S, \mathcal{T})$, i.e., the large dataset is generally more similar to the test than the small dataset. We aim to identify a maximally large subset $\tilde{\mathcal{D}}_L \subseteq \mathcal{D}_L$ of the large training set, such that its similarity gap $S(\tilde{\mathcal{D}}_L, \mathcal{T})$ is equal to the similarity gap $S(\mathcal{D}_S, \mathcal{T})$ of the small dataset (on a per-sample basis, meaning $s_i(\tilde{\mathcal{D}}_S) = s_i(\mathcal{D}_S)$ for all samples). To achieve this, we examine each test sample $t_i$ and remove any sample $x \in \mathcal{D}_L$ for which the similarity $s(t_i, x) > s_i(\mathcal{D}_S)$. We illustrate this procedure in Fig. 5.

This method allows us to surgically remove *highly similar images* with respect to a given test set and reference training set. Compared to the unconstrained pruning in Sec. 4.2, this will remove far less samples from LAION-400M, and thus allows us to isolate the impact of *highly similar images*.

## 5 CORRECTING FOR HIGHLY SIMILAR IMAGES

We now apply the framework from Sec. 4.4 to remove highly similar images from LAION-200M. To ensure that ImageNet-Train and LAION-200M have the same similarity gap to the test sets, we include all ImageNet-Train images in LAION-200M with the caption "a photo of a {object class}". We refer the reader to Appx. Sec. C for a discussion on the choice of ImageNet for our experiments.

Table 1: **Corrected zero-shot performance of CLIP ViT-B/32.** 'X-pruned' represents a pruned dataset from LAION-200M + ImageNet such that the similarity gap to 'X' is the same as the similarity gap of ImageNet to 'X'. The sizes of these subsets are subtracted from the LAION-200M + ImageNet's size. Here, 'X' is one of the six standard ImageNet test sets. 'combined-pruned' splits ensure a similarity gap of LAION-200M and ImageNet-Train to all 6 test sets. CLIP's corrected zero-shot performance drops the most on ImageNet-Sketch and ImageNet-R with a relative performance drop of $10.8\,\%$ and $4.8\,\%$ respectively. Red color indicates a drop in performance on the respective test set, and blue represents a rise. Overall, high performance indicates that highly similar images do not play a key role in explaining CLIP's generalization ability.

| Dataset | Size | Top-1 Accuracy | | | | | |
| | | Val | Sketch | A | R | V2 | ON |
|---|---|---|---|---|---|---|---|
| OpenAI (Radford et al., 2021) | 400 000 000 | 63.38 | 42.32 | 31.44 | 69.24 | 55.96 | 44.14 |
| L-400M (Schuhmann et al., 2021) | 413 000 000 | 62.94 | 49.39 | 21.64 | 73.48 | 55.14 | 43.94 |
| L-200M | 199 824 274 | 62.12 | 48.61 | 21.68 | 72.63 | 54.16 | 44.80 |
| L-200M + IN-Train | 200 966 589 | 68.66 | 50.21 | 23.33 | 72.9 | 59.7 | 43.99 |
| — val-pruned | −377 340 | 68.62 | 49.58 | 23.47 | 72.74 | 59.47 | 45.08 |
| — sketch-pruned | −8 342 783 | 68.34 | 44.78 | 22.7 | 69.35 | 59.52 | 44.12 |
| — a-pruned | −138 852 | 68.85 | 50.25 | 22.99 | 72.44 | 60.05 | 44.43 |
| — r-pruned | −5 735 749 | 68.71 | 46.92 | 23.44 | 69.48 | 59.6 | 45.08 |
| — v2-pruned | −274 325 | 68.79 | 50.45 | 23.19 | 72.58 | 59.84 | 45.33 |
| — objectnet-pruned | −266 025 | 68.75 | 50.14 | 22.70 | 72.82 | 59.37 | 43.73 |
| — combined-pruned | −12 352 759 | 68.05 | 44.12 | 22.15 | 67.88 | 58.61 | 44.39 |

As described in Sec. 4.4, we first compute the similarity gaps of the smaller dataset, i.e., ImageNet-Train, to the samples in each of the six test sets. Pruning LAION-200M to these similarity gaps leaves us with six different base splits as shown in Tab. 1. We also generate a 'combined-pruned' split that ensures an ImageNet-Train-like similarity gap to all test sets simultaneously. We can now train CLIP from scratch on these splits to obtain a corrected zero-shot performance and compare it to the accuracy of CLIP trained by OpenAI and OpenClip (Ilharco et al., 2021; Radford et al., 2021).

The first important point to note in Tab. 1 is that for 'sketch-pruned' and 'r-pruned' datasets, we prune 8.3M and 5.7M samples, respectively. For all other datasets, we prune only around 250K-380K samples. We saw indications of this already in Sec. 4 when we looked at the distribution of nearest-neighbor similarities, see also Tab. 7. The number of pruned samples is also highly correlated with the respective accuracies. For CLIP trained on the 'r-pruned' dataset and CLIP trained on the 'sketch-pruned' dataset, we observe a $4.8\,\%$ relative performance decrease on ImageNet-R and $10.8\,\%$ relative performance decrease on ImageNet-Sketch compared to the baseline. There is also a considerable performance change on ImageNet-R for 'sketch-pruned' and on ImageNet-Sketch for 'r-pruned'. This is reasonable as there is some style overlap in ImageNet-Sketch and ImageNet-R. For the other four base splits, we see less than $1\,\%$ relative performance change on all six evaluation sets. The performance of the CLIP model trained on the 'combined-pruned' split is lower than the baseline on all six eval sets, with sizeable drops in ImageNet-R and ImageNet-Sketch. We also observe similar trends when we do not add ImageNet-Train to the pruned datasets (refer to Tab. 4 in the Appx.).

## 6 DISCUSSION

We now return to our original question: *Does CLIP's accuracy on OOD benchmarks mainly stem from highly similar images in its train set?* To give a definitive answer, we take a closer look at the CLIP model trained on 'sketch-pruned'. This model's training set is as dissimilar to ImageNet-Sketch as is ImageNet-Train. It features an accuracy of $68.34\,\%$ on ImageNet-Val. According to ImageNet-Train's *effective robustness line* (Fang et al., 2022), at this performance level, we would expect an accuracy of roughly $14\,\%$ on ImageNet-Sketch. Instead, we find an accuracy of $44.78\,\%$. In other words, training on a much larger dataset while keeping the similarity gap constant drastically increases generalization performance for CLIP (in this case, by a staggering 30 percentage points). This effect is even higher for other datasets. *This indicates that CLIP's impressive performance is not so much the result of*

*a high train-test similarity but that CLIP leverages its dataset scale and diversity to learn more generalizable features.*

**What drives generalization?** Generalization of vision-language models is a complex subject where several factors like architectural choices, caption quality, training procedures, and data distribution play a role. We focus on the training distribution since prior works have studied the effect of the aforementioned factors on CLIP's generalization performance (e.g., Santurkar et al., 2022; Mintun et al., 2021) and identified it as a prominent factor (Fang et al., 2022). Many distribution properties could contribute to generalization performance, but based on raw visualizations of the involved datasets, highly similar images are clearly *a* factor. Our results only show that it is not the most salient factor and a large chunk of performance remains to be explained. We leave the scrutiny of other likely factors like data diversity and density for future work. Our work should be interpreted as a step towards finding specific data properties that dictate generalization.

**Measuring the true OOD performance** Our analysis excluded training images from LAION with a smaller similarity gap to test images compared to ImageNet Train. Another interesting analysis would be to prune LAION images to measure its true OOD performance. To remove all images of a certain domain, we need to be able to label each image as 'ID' or 'OOD'. This essentially means that we need access to a domain classifier (which would also need near-perfect accuracy so that no images are overlooked). Even for the 'sketch' domain, where a classifier could conceivably be trained, it is unclear exactly how the classifier should demarcate this domain: Should the domain contain all sketches, even sketches with characteristics not present in ImageNet-Sketch? What about tattoos or small sketches on objects in natural images? For other benchmarks, such as ImageNet-A, it is even less clear how the test images constitute a well-separable domain of images. This vagueness in defining a domain based on a given test set prevents us from building a fair OOD setting, which is why we do not analyze or claim to analyze this.

**Similarity metric** We defer the reader to Sec. E for a discussion and ablation on the choice of CLIP ViT-B/16+ as the similarity metric.

**Highly similar images** We want to clarify further the notion of *highly similar images*. In Secs. 4.1, 4.2, and 4.3, when we use the notion of *similar images* to a given image sample, we refer to images with high perceptual similarity values with no precise constraint. In contrast, in Secs. 4.4 and 5 we impose a constraint that defines *highly similar images* to a sample as images that are closer to LAION-200M than ImageNet-Train based on our perceptual similarity metric.

**Does compositionality drive performance?** In this work, we found that high train-test similarity is insufficient to explain CLIP's high generalization performance on OOD test sets. In our analysis, we only excluded images that were highly similar to the training set to maintain the same similarity gap with respect to ImageNet Train, e.g. sketches of dogs if the test image was a sketch of a dog. However, sketches of other animals and objects still remained in CLIP's training set. An open question remains whether compositionality (Wiedemer et al., 2023) can close the gap between the object and its domain, i.e. whether CLIP can generalize from sketches of cats and natural images of dogs to understanding sketches of dogs.

## 7  CONCLUSION

CLIP has demonstrated unprecedented performance on common OOD benchmarks designed originally for ImageNet. Given that the training dataset of CLIP is so large and diverse, it is natural to wonder whether its performance stems from the sheer similarity of many training samples to the benchmarks. To the best of our knowledge, we are the first to systematically test if high train-test similarity dictates CLIP's generalization performance. In our work, we address this by pruning away samples from the training set that are more similar to the test sets than ImageNet samples. Models trained on the pruned dataset do not significantly lose performance and still exhibit stellar generalization capabilities far beyond performance-matched ImageNet-trained models. This indicates that high similarity to the test sets alone can not explain CLIP's generalization ability. We hope this result will prompt the community to investigate other factors that allow models to learn more generalizable features from web-scale datasets.

## REPRODUCIBILITY STATEMENT

For all the basic details of training, pruning, similarity computation, and other analysis, we defer the reader to Sec. 3. Details of computing the similarities and its correlation to accuracy is given in the caption of Figs. 2, 3, and Sec. 4.1. To perform the experiment that observes the effect of 'near-pruning' and 'far-pruning', we defer the reader to Sec. 4.2 and the caption of Fig. 3. The core methodology of our paper is clearly elucidated in Section 4.4. Furthermore, the details of generating the datasets and training the models are given in the first and second paragraph of Sec. 5, and in the caption of Tab. 1.

## AUTHOR CONTRIBUTIONS

The project was led and coordinated by PM. The method was jointly developed by PM, TW, with insights from ER, WB, MB. PM conducted all the experiments based on code jointly implemented by PM and TW. PM, TW, ER, and WB jointly wrote the manuscript with additional insights from MB. ER created all figures and visualizations with TW's help using data provided by PM and with comments from WB.

## ACKNOWLEDGMENTS

We would like to thank (in alphabetical order): Thomas Klein, George Pachitariu, Matthias Tangemann, Vishaal Udandarao, Max Wolff, and Roland Zimmermann for helpful discussions, feedback, and support with setting up the experiments. This work was supported by the German Federal Ministry of Education and Research (BMBF): Tübingen AI Center, FKZ: 01IS18039A. WB acknowledges financial support via an Emmy Noether Grant funded by the German Research Foundation (DFG) under grant no. BR 6382/1-1 and via the Open Philantropy Foundation funded by the Good Ventures Foundation. WB is a member of the Machine Learning Cluster of Excellence, EXC number 2064/1 – Project number 390727645. This research utilized compute resources at the Tübingen Machine Learning Cloud, DFG FKZ INST 37/1057-1 FUGG. We thank the International Max Planck Research School for Intelligent Systems (IMPRS-IS) for supporting PM, TW, and ER.

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

## A    DISTRIBUTIONAL DISSIMILARITIES OF LAION-200M AND IMAGENET

### A.1    NEAREST-NEIGHBOR SIMILARITY BETWEEN LAION / IMAGENET-TRAIN AND OTHER OOD DATASETS

As an extension of our analysis in Sec. 4, we plot the nearest-neighbor similarity between ImageNet-Train/LAION-400M and other OOD test sets, namely ImageNet-A (Hendrycks et al., 2021), Object-Net (Barbu et al., 2019) and ImageNet-V2 (Recht et al., 2019), and display our results in Figure 6. There are no significant differences in nearest-neighbor similarity for these test sets. Similar to our results in Figure 3 (left), we find a strong correlation between perceptual similarity to LAION-400M and the top-1 accuracy of our LAION-trained model.

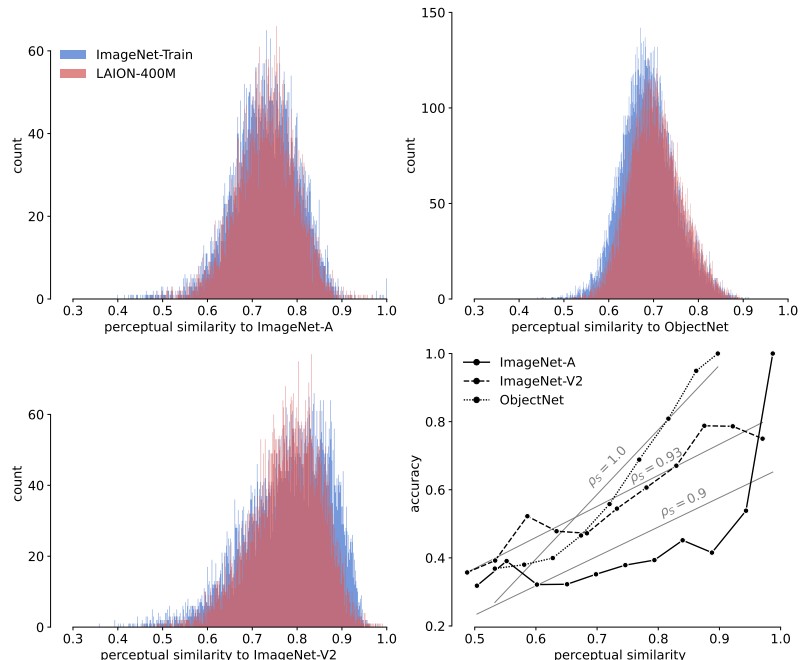

Figure 6: **Similarity of nearest neighbors to test sets varies between LAION-400M and ImageNet-Train and is correlated with performance.** Histograms over the nearest-neighbor similarity of test sets ImageNet-A (top left), ObjectNet (top right), and ImageNet-V2 (bottom left) to training sets LAION-400M (red) and ImageNet-Train (blue). There are no significant differences in nearest-neighbor similarity for these test sets. Nearest-neighbor similarity of test points to LAION-400M samples and top-1 classification accuracy is strongly correlated (bottom right). Data points in the correlation plot are averaged over bins (interval = 0.05) of the red histograms.

### A.2    DUPLICATES

We do a duplicate analysis in Tab. 2. To estimate the number of test points with *near duplicates*, we project the test set and LAION-400M to CLIP's image embedding space and check if any point in LAION-400M lies in the vicinity ($\epsilon = 0.05$) of each of the query test points.

## B    ADDITIONAL EXPERIMENTAL RESULTS

### B.1    IMPACT OF NEAR/FAR-PRUNING ON ALL DATASETS

In Sec. 4.2, by using each of the test datasets ImageNet-Val and ImageNet-Sketch and near/far-pruning LAION-200M, we trained models and reported the performance on the test datasets, respectively. We now plot the performance of these models on all six datasets in Figure 7. 'Near-pruning' ('far-pruning') with ImageNet-Sketch results in lower (higher) performance than 'near-pruning'

Table 2: **Number of test points of OOD datasets for which we find *near duplicates* in ImageNet-Train and LAION-400M**. A data point is considered *near duplicate (semantic duplicate)* if the distance in the CLIP embedding space is less than 0.05 (Abbas et al., 2023).

| | | Duplicates | |
|---|---|---|---|
| **Dataset** | **Size** | **ImageNet-Train** | **LAION-400M** |
| ImageNet-Val | 50000 | 1336 | 70 |
| ImageNet-Sketch | 50889 | 18 | 1553 |
| ImageNet-R | 30000 | 104 | 297 |
| ImageNet-A | 7500 | 10 | 5 |
| ImageNet-V2 | 10000 | 10 | 24 |
| ObjectNet | 18574 | 0 | 0 |

('far-pruning') with ImageNet-Val on ImageNet-R and ImageNet-Sketch. Likewise, 'near-pruning' ('far-pruning') with ImageNet-Val results in lower (higher) performance than 'near-pruning' ('far-pruning') with ImageNet-Sketch on ImageNet-Val, ImageNet-V2, and ObjectNet. This is expected because ImageNet-Sketch is characteristically closer to ImageNet-R, and ImageNet-Val is closer to ImageNet-V2, ObjectNet, and ImageNet-A.

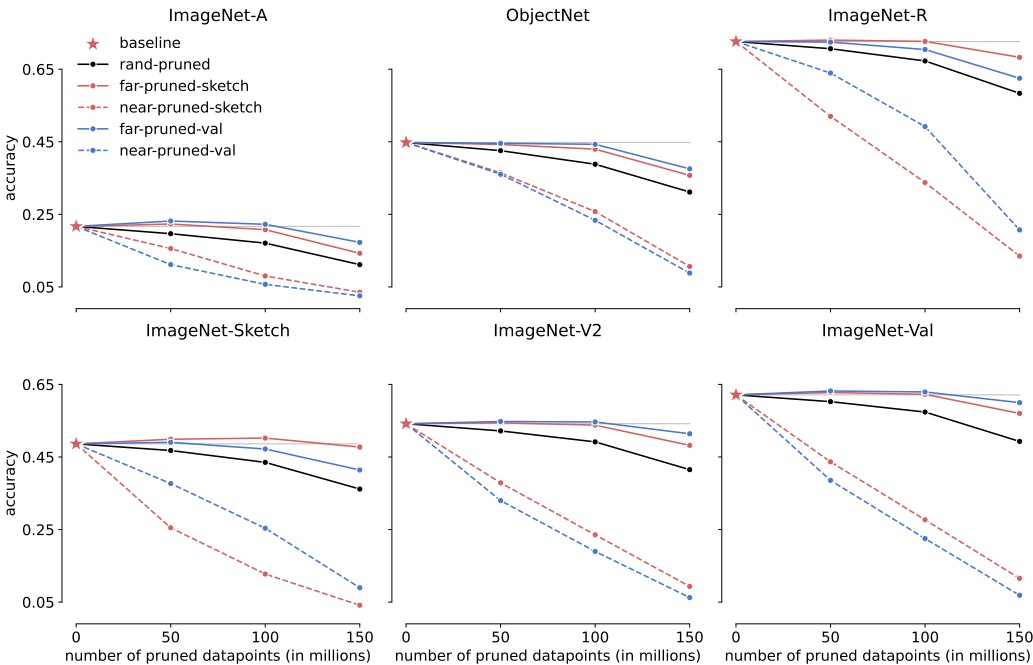

Figure 7: **The effect of 'near-pruning' and 'far-pruning' with ImageNet-Sketch or ImageNet-Val as the query dataset on the performance of all six test sets.** CLIP's zero-shot accuracy as a function of the number of pruned points from LAION-200M. The baseline model is trained on de-duplicated LAION-400M, which we call LAION-200M. To generate the 'near-pruned' datasets, we remove the images in the decreasing order of similarity (based on CLIP image-embedding similarity) to each of the test sets ImageNet-Sketch and ImageNet-Val, respectively. In contrast, the 'far-pruned' datasets are generated by dropping images in the increasing order of similarity values to the respective test sets. For the 'rand-pruned' datasets, we prune random points.

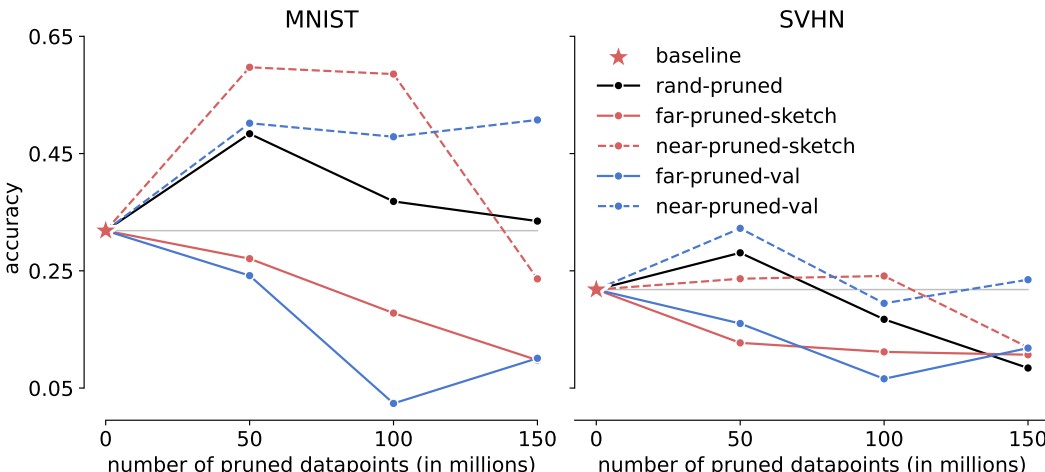

Figure 8: **The effect of 'near-pruning' and 'far-pruning' with ImageNet-Sketch or ImageNet-Val as the query dataset on the performance on MNIST / SVHN** CLIP's zero-shot accuracy as a function of the number of pruned points from LAION-200M. The baseline model is trained on deduplicated LAION-400M, which we call LAION-200M. To generate the 'near-pruned' datasets, we remove the images in the decreasing order of similarity (based on CLIP image-embedding similarity) to each of the test sets ImageNet-Sketch and ImageNet-Val, respectively. In contrast, the 'far-pruned' datasets are generated by dropping images in the increasing order of similarity values to the respective test sets. For the 'rand-pruned' datasets, we prune random points.

### B.2    IMPACT OF NEAR/FAR PRUNING ON NON-IMAGENET-LIKE DATASETS

In Fig. 7, we observe a consistent trend across all datasets that near-pruning with respect to either ImageNet-Sketch or ImageNet-Val decreases performance while performance is stable (at times even increases) when doing far-pruning. These findings can be explained by two hypotheses:

1. The pruned images in the near-pruning setting are reasonably similar to ImageNet-Sketch or ImageNet-Val, thus we see a drop in performance when we train CLIP on the pruned datasets.

2. Near-pruning with respect to ImageNet-Sketch or ImageNet-Val results in pruning datapoints that are of the highest quality samples from LAION which will perform well on any downstream task, when trained upon.

To decide between these two hypotheses, we repeat the analysis on two test sets which are very dissimilar from ImageNet: SVHN (Netzer et al., 2011) and MNIST (Deng, 2012). We show the results in Fig. 8 and indeed observe a reversed trend compared to Fig. 7: Now, for a several cases, near-pruning increases performance and far-pruning decreases it, respectively. Thus, the near-pruned datapoints are **not** comprised of high quality samples which improve performance on all downstream tasks, and pruning away images similar to either ImageNet-Sketch or ImageNet-Val only decreases performance on ImageNet-like datasets.

### B.3    NEAR/FAR PRUNING EXPERIMENTS ON IMAGENET-TRAIN

We generate new datasets by near/far pruning datapoints on ImageNet-Train with test datasets ImageNet-Sketch and ImageNet-Val. The similarities are computed in the pre-trained CLIP ViT-B/16+ embedding space. We then train ResNet18s until convergence using standard PyTorch hyperparameter settings. We report the absolute and relative performance (to the respective baseline) in Figure 9 and  10. We observe that near/far pruning affects CLIP performance more than ResNet18.

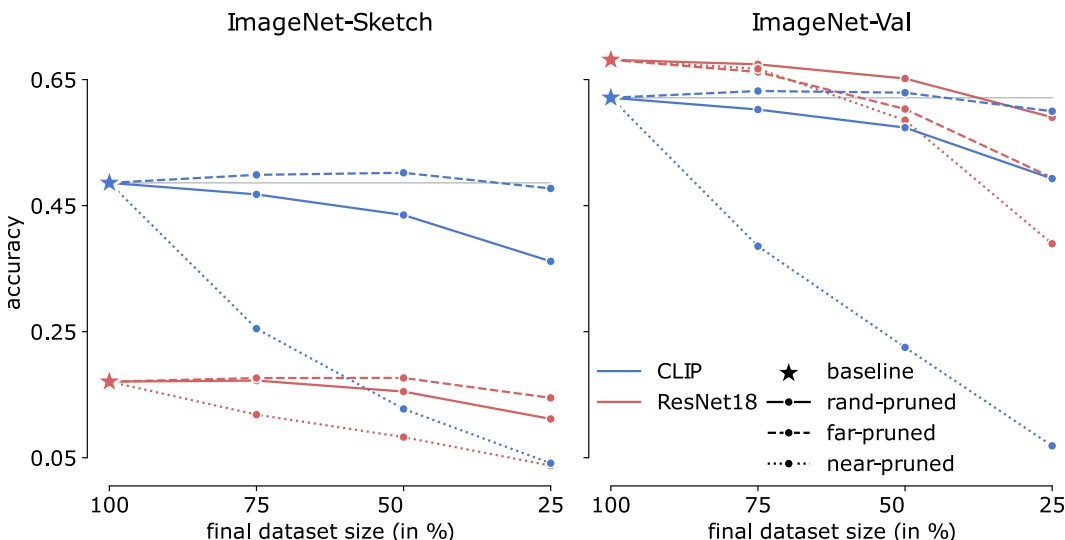

Figure 9: **Effect of pruning similar and dissimilar points to a given test set on both CLIP and ResNet performance**. The baseline model of CLIP is trained on de-duplicated LAION-400M, which we call LAION-200M. The baseline model of ResNet18 is trained on ImageNet-Train. To generate the 'near-pruned' datasets, we remove images in decreasing order of similarity to ImageNet-Sketch or ImageNet-Val (based on CLIP image-embedding similarity). In contrast, the 'far-pruned' datasets are generated by pruning images in the increasing order of similarity values to the respective test sets. For the 'rand-pruned' datasets, we prune random points. Pruning similar images adversely affects performance compared to pruning dissimilar or random images. Generally, near/far pruning affects CLIP more than ResNet18.

### B.4    CORE SET OF 100M

In Sec. 4.2, we identify a 100M core set of LAION-400M, which, when trained on, leads to a CLIP model that nearly matches the performance of a LAION-400M trained CLIP model on the six test datasets. Motivated by the performance increase of the 'far-pruning' technique in the previous results, we now build several core sets of 100M, which, when trained on, roughly match the performance of CLIP trained on LAION-400M. Instead of pruning from the farthest point to samples in just a single test set in CLIP ViT-B/16+'s embedding space, we now prune from the farthest point to samples from a collection of test sets (all six ImageNet-1k OOD test sets). We do far-pruning with all of the test sets on both LAION-200M and LAION-400M to obtain datasets that we call 'all-far-pruned'. For comparison, we also add the performance of CLIP trained on far-pruned datasets with query datasets as ImageNet-Sketch and ImageNet-Val, which we call 'sketch-far-pruned' and 'val-far-pruned', respectively.

We report the results in Tab. 3 and observe that models trained on all of the splits are within 3 % average accuracy range of CLIP trained on LAION-400M. The model with the highest average accuracy is trained on 'all-far-pruned (L-200M)', which is a dataset generated by pruning far or dissimilar images in LAION-200M with all 6 test datasets as query datasets. This model also performs better than a model trained on a dataset of the same size generated by the pruning technique SemDeDup (Abbas et al., 2023). SemDeDup aims to prune semantically similar data with minor loss in test performance. We do not suggest this coreset as an alternative to other pruning or deduplication methods that are largely agnostic to the downstream test datasets. Instead, we here created a coreset that is specifically designed to perform well on six OOD test sets to facilitate further research into what aspects drive generalization.

### B.5    MAIN EXPERIMENTS WITHOUT ADDING IMAGENET-TRAIN

We repeat the experiments in Sec. 5 without adding ImageNet-Train to LAION-200M and report results in Tab. 4. We observe the same trends as in Tab. 1.

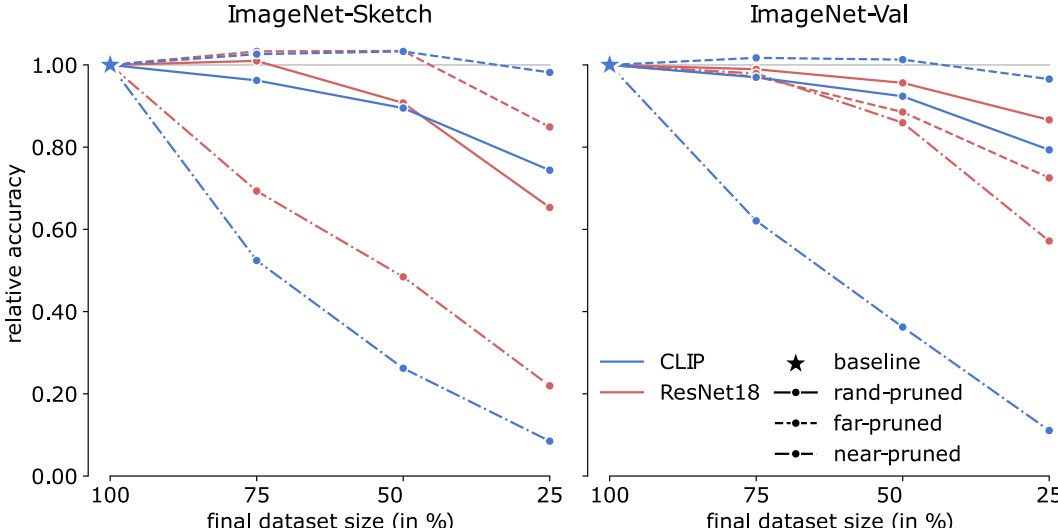

Figure 10: **Effect of pruning similar and dissimilar points to a given test set on both CLIP and ResNet's performance relative to the baseline**. The baseline model of CLIP is trained on de-duplicated LAION-400M, which we call LAION-200M. The baseline model of ResNet18 is trained on ImageNet-Train. To generate the 'near-pruned' datasets, we remove images in decreasing order of similarity to ImageNet-Sketch or ImageNet-Val (based on CLIP image-embedding similarity). In contrast, the 'far-pruned' datasets are generated by pruning images in the increasing order of similarity values to the respective test sets. For the 'rand-pruned' datasets, we prune random points. Pruning similar images adversely affects performance compared to pruning dissimilar or random images. The relative performance drop curves indicate that near/far pruning affects CLIP more than ResNet18.

Table 3: **Performance of 'far-pruned' CLIP (ViT-B/32) on the six test sets.** We do 'far-pruning' on LAION-400M with all 6 test sets as query sets and obtain the dataset 'all-far-pruned (L-400M).' Similarly, we do 'far-pruning' on LAION-400M with all with all 6 test sets as query sets, ImageNet-Sketch, and ImageNet-Val to get the datasets 'all-far-pruned (L-200M)', 'sketch-far-pruned (L-200M)', and 'val-far-pruned (L-200M)' respectively. These models are compared to off the shelf CLIP model (Ilharco et al., 2021), model trained on LAION-200M, and a model trained on SemDeDup (Abbas et al., 2023) dataset of size 100M.

| Dataset | Size | Top-1 Accuracy | | | | | | |
| --- | --- | --- | --- | --- | --- | --- | --- | --- |
| | | Val | Sketch | A | R | V2 | ON | Avg. |
| L-400M | 400M | 62.94 | 49.39 | 21.64 | 73.48 | 55.14 | 43.94 | *51.09* |
| L-200M | 199.8M | 62.12 | 48.61 | 21.68 | 72.63 | 54.16 | 44.80 | *50.67* |
| all-far-pruned (L-400M) | 100M | 61.90 | 48.11 | 19.43 | 70.14 | 53.11 | 39.30 | *48.67* |
| all-far-pruned (L-200M) | 100M | 62.80 | 49.23 | 21.6 | 72.3 | 54.72 | 43.64 | *50.71* |
| val-far-pruned (L-200M) | 100M | 62.79 | 47.53 | 21.65 | 70.40 | 54.35 | 43.70 | *50.07* |
| sketch-far-pruned (L-200M) | 100M | 62.27 | 50.21 | 20.77 | 72.67 | 53.77 | 42.95 | *50.44* |
| SemDeDup | 100M | 52.19 | 41.70 | 16.71 | 67.05 | 44.96 | 39.59 | *43.7* |

## C  ON THE CHOICE OF IMAGENET

We choose zero-shot classification on ImageNet and its distribution shifts as the main object of study four our work. Our analysis is agnostic to these choices and one could potentially use other datasets like iWILDCam 2021 (Beery et al., 2021), FMoW (Christie et al., 2018), MS-COCO (Lin et al., 2015) and Flickr30k (Yonglong Tian et al., 2021) and on tasks like image retrieval. One reason why we

Table 4: **Corrected zero-shot performance of CLIP ViT-B/32.** 'X-pruned' represents a pruned dataset from LAION-200M such that the similarity gap to 'X' is roughly the same as the similarity gap of ImageNet to 'X'. The sizes of these subsets are subtracted from the LAION-200M's size. Here, 'X' is one of the six standard ImageNet test sets. 'combined-pruned' splits ensure a similarity gap of LAION-200M and ImageNet-Train to all 6 test sets. CLIP's corrected zero-shot performance drops the most on ImageNet-Sketch and ImageNet-R with a relative performance drop of $11.08\%$ and $5.99\%$ respectively. Red color indicates a drop in performance on the respective test set. Overall, high performance indicates that highly similar images do not play a key role in explaining CLIP's generalization ability.

| | | | Top-1 Accuracy | | | | | |
|---|---|---|---|---|---|---|---|---|
| **Model** | **Dataset** | **Size** | **Val** | **Sketch** | **A** | **R** | **V2** | **ObjectNet** |
| ViT-B/32 | OpenAI | 400 000 000 | 63.38 | 42.32 | 31.44 | 69.24 | 55.96 | 44.14 |
| ViT-B/32 | L-400M | 413 000 000 | 62.94 | 49.39 | 21.64 | 73.48 | 55.14 | 43.94 |
| ViT-B/32 | L-200M | 199 824 274 | 62.12 | 48.61 | 21.68 | 72.63 | 54.16 | 44.80 |
| ViT-B/32 | ⌐ val-pruned | −377 340 | 62.12 | 48.38 | 21.45 | 72.2 | 54.76 | 42.79 |
| ViT-B/32 | ⌐ sketch-pruned | −8 342 783 | 61.55 | 43.22 | 22.28 | 69.6 | 53.53 | 42.77 |
| ViT-B/32 | ⌐ a-pruned | −138 852 | 62.49 | 48.49 | 21.63 | 72.15 | 54.38 | 43.25 |
| ViT-B/32 | ⌐ r-pruned | −5 735 749 | 61.73 | 45.66 | 21.67 | 68.28 | 54.1 | 42.90 |
| ViT-B/32 | ⌐ v2-pruned | −274 325 | 62.48 | 48.62 | 22.13 | 72.3 | 53.83 | 43.38 |
| ViT-B/32 | ⌐ objectnet-pruned | −266 025 | 62.30 | 49.03 | 22.64 | 72.90 | 54.21 | 42.80 |
| ViT-B/32 | ⌐ combined-pruned | −12 352 759 | 61.5 | 41.97 | 21.72 | 67.25 | 53.65 | 42.23 |
| ResNet-101 | ImageNet-1k | 1 200 000 | 77.21 | 27.58 | 4.47 | 39.81 | 65.56 | 36.63 |

do not investigate this in our paper for the reason that several ImageNet distribution shifts are more human aligned than for the aforementioned datasets. For instance, there's a perceptual demarcation between a sketch of a dog (ImageNet-Sketch) and a natural image of a dog (ImageNet-Train). In contrast, it is unclear what the distribution shift of Flickr-30k is from MS-COCO. Additionally, retrieval tasks are more complex and highly sensitive to captions, demanding an analysis that factors in both images and texts. Another reason to choose zero-shot classification and ImageNet is that CLIP demonstrated unprecedented performance on ImageNet-based distribution shifts (Radford et al., 2021). Moreover, we find iWILDCam and FMoW datasets problematic since CLIP's (ViT-B/32) zero-shot performance on them is rather low (7.45% and 12.96%) (Ilharco et al., 2021). We therefore chose ImageNet and its distribution shifts for our study and leave analysis on other datasets for future work.

## D SIMILARITY ANALYSIS ON CELEBA AND WATERBIRDS

Secs. 4 and 5 only considered test sets with a clear distribution shift with respect to ImageNet-Train. However, the general method outlined in Sec. 4.4 is dataset-agnostic. To illustrate this point, we here consider test sets that exhibit distribution shifts with respect to other datasets. Specifically, we consider CelebA (Liu et al., 2015) and Waterbirds Sagawa et al. (2019).

**CelebA** This dataset contains 202 599 celebrity images with 40 annotated attributes (Liu et al., 2015). Gannamaneni et al. (2023) showed that CLIP could zero-shot predict many attributes with high accuracy (between $53\%$ and $97\%$ top-1 accuracy). We can split the data along each attribute to obtain training and test sets with a specific distribution shift. In Fig. 11, we repeat the similarity analysis from Sec. 4.3 for CelebA splits along the 'eyeglasses' and 'hat' attributes. The distribution of nearest-neighbor similarities to the test set CelebA-w/-eyeglasses (CelebA-w/-hat) differs between LAION-400M and CelebA-w/o-eyeglasses (CelebA-w/o-hat). We again observe a strong correlation between the similarity of a test sample to LAION-400M and CLIP's zero-shot accuracy in predicting the 'gender' attribute.

**Waterbirds** We generate two sets of splits of this dataset. We first split by background (land or water) and obtain a distribution shift from Waterbirds-land with 7051 images (6220 landbirds, 831

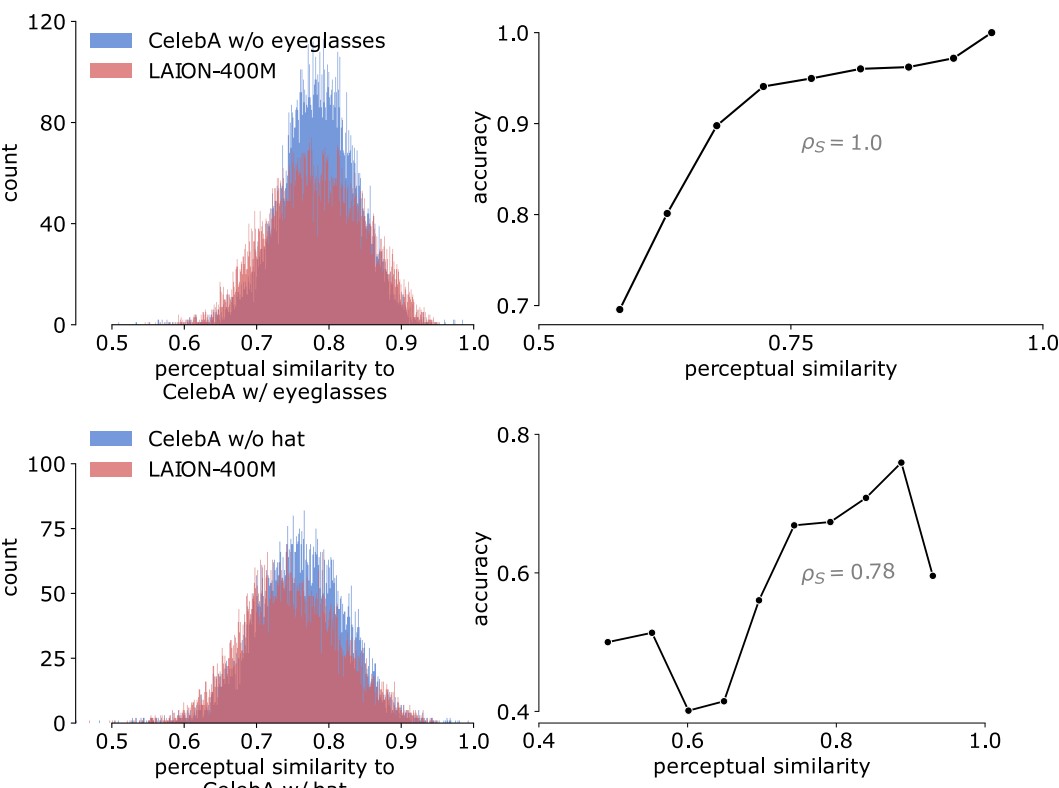

Figure 11: **nearest-neighbor similarity distribution and correlation to zero-shot accuracy for CelebA**. **Left**: The histogram shows the similarity of samples in CelebA-w/-eyeglasses to their nearest neighbors in LAION-400M (red) and CelebA-w/o-eyeglasses (blue). **Right**: The strong correlation between perceptual similarity of test points to nearest neighbors in LAION-400M samples and CLIP's top-1 classification accuracy on Male/Female classification indicates that differences in similarity can be expected to impact the performance of LAION-trained models. Data points in the correlation plot are averaged over bins (interval = 0.05) of the red histograms in the left plot.

waterbirds) to Waterbirds-water with 4737 images (2905 landbirds, 1832 waterbirds). We then split into the core group and the worst group. The core group consists of 8052 images of landbirds on land or waterbirds on water. The worst group consists of 3736 images of landbirds on water or waterbirds on land. In Fig. 12, we repeat the similarity analysis from Sec. 4.3 for Waterbirds splits along the land/water and core/worst-group distribution shifts. The distribution of nearest-neighbor similarities to the test set Waterbirds-water (Waterbirds-worst group) differs between LAION-400M and Waterbirds-land (Waterbirds-core group), and we again observe a strong correlation between the similarity of a test sample to LAION-400M and CLIP's zero-shot accuracy in predicting the landbird/waterbird class.

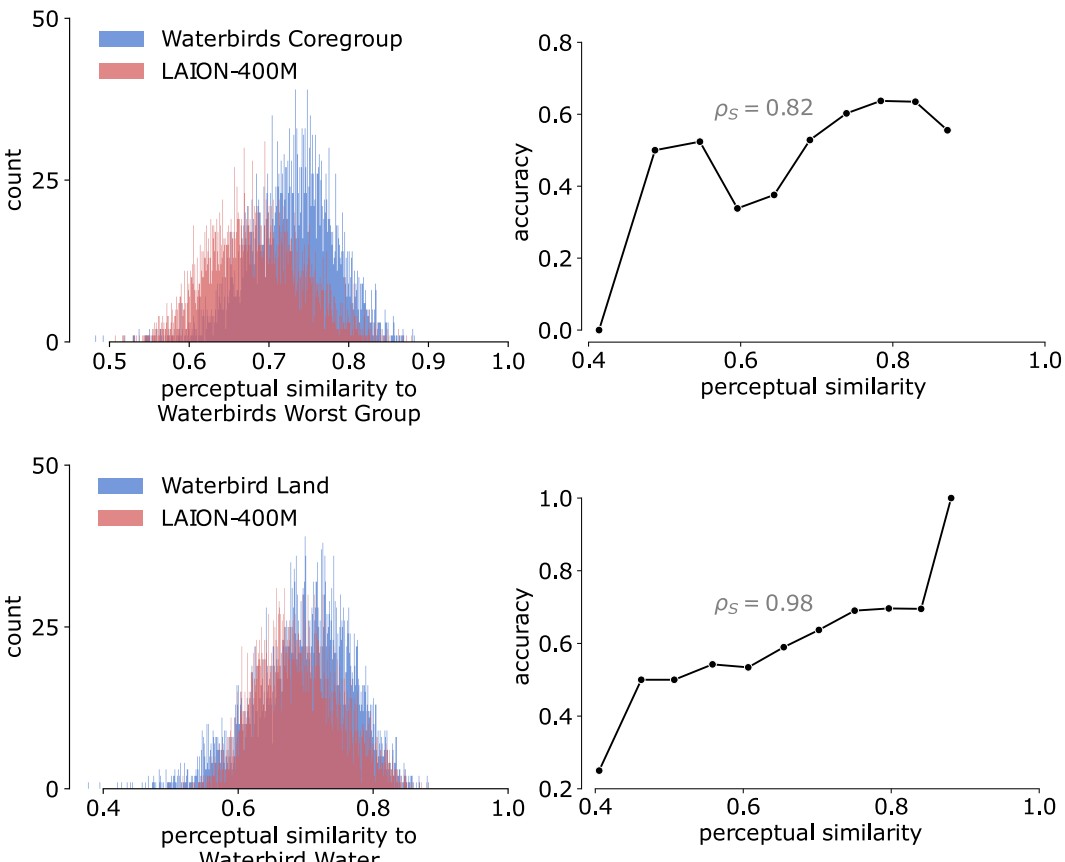

Figure 12: **nearest-neighbor similarity distribution and correlation to zero-shot accuracy for Waterbirds**. **Left**: The histogram shows the similarity of samples in Waterbirds-water to their nearest neighbors in LAION-400M (red) and Waterbirds-land (blue). **Right**: The strong correlation between perceptual similarity of test points to nearest neighbors in LAION-400M samples and CLIP's top-1 classification accuracy on landbird/waterbird classification indicates that differences in similarity can be expected to impact the performance of LAION-trained models. Data points in the correlation plot are averaged over bins (interval = 0.05) of the red histograms in the left plot.

## E    COMPARING EMBEDDING METRICS

To our knowledge, *perceptual similarity* as measured in CLIP ViT-B/16+'s image embedding space is a leading metric to capture the semantic and stylistic similarity between images. While we found this metric to align well with our intuitive notion of similarity and believe it to have captured the vast majority of highly similar images (see also Appx. G.2 where we visualize the pruned datasets), we cannot guarantee that all highly similar images were removed. While we believe, based on prior work (Fu et al., 2023; Abbas et al., 2023; Gadre et al., 2023; Zhang et al., 2021) and our analysis, that CLIP's embedding space sufficiently captures relevant features, in this section, we ablate the influence of the embeddings used to compute the perceptual similarity. Specifically, we compare CLIP ViT-B/16+ embeddings used throughout the main paper to the embeddings of ViT-L-14 trained on LAION-400M, a much larger model.

We compute the nearest-neighbor similarities of ImageNet-Train to the test sets using either embeddings and compute their correlation. We summarize the results in table 5 and find that the correlation is strong across test sets. Figure 13 also shows histograms of nearest-neighbor similarities using either embedding, revealing that the similarity distributions are very similar. Given these two comparisons, and considering that CLIP ViT-B/16+ embeddings are faster and cheaper to compute and have been shown to capture perceptual image similarities reasonably well by previous work (Abbas et al., 2023), we use them throughout our work.

Table 5: **Choice of embeddings has little impact on nearest-neighbor similarities**. We compute nearest-neighbor similarities between ImageNet-Train the six tests on embeddings produced by CLIP ViT-B/16+ and ViT-L-14 and find a strong correlation across the bench.

| Dataset | $\rho_S$ |
|---|---|
| ImageNet-Val | 0.93 |
| ImageNet-Sketch | 0.87 |
| ImageNet-R | 0.90 |
| ImageNet-A | 0.86 |
| ImageNet-V2 | 0.93 |
| ObjectNet | 0.80 |

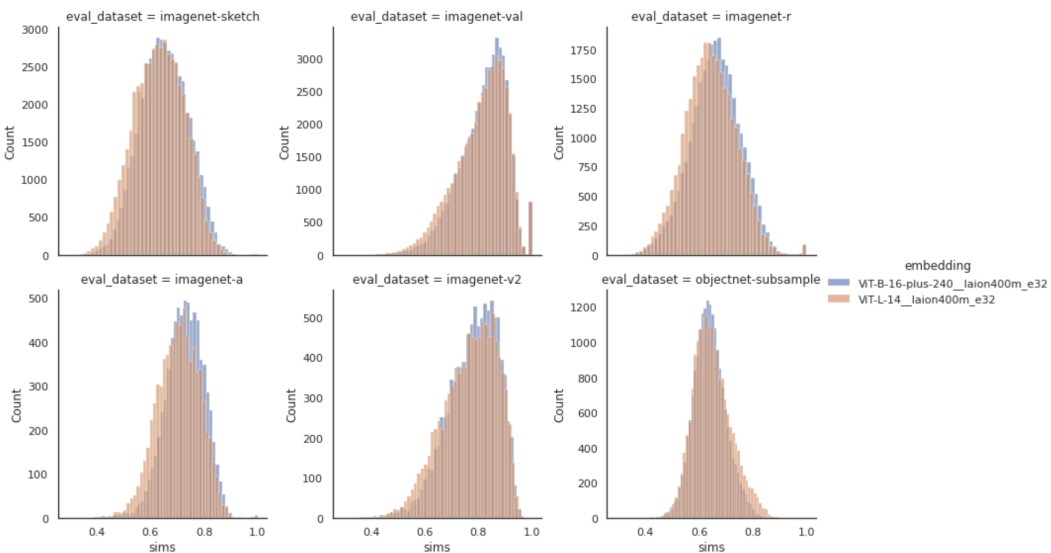

Figure 13: **Choice of embeddings has little impact on nearest-neighbor similarity distribution**. We compute nearest-neighbor similarities between ImageNet-Train the six tests on embeddings produced by CLIP ViT-B/16+ and ViT-L-14 and find their distributions visually very similar across the bench.

# F    DISTRIBUTION OF SIMILARITIES OF LAION-200M AND IMAGENET-TRAIN AFTER PRUNING

We analyzed nearest-neighbor similarity distribution of the test sets to LAION-200M and ImageNet-Train (see Figs. 4 and 6). But what about the nearest-neighbor similarity distributions of LAION-200M and ImageNet-Train to the test sets, especially after pruning? We now answer this question to understand better where the training points are situated with respect to the pruning boundary.

For each data point in the pruned dataset, we compute the maximum similarity to the respective test set and divide it by the test point's similarity gap (i.e., nearest-neighbor similarity of the test point to LAION-200M before pruning). We call this quantity *normalized similarity*. Note that the normalized similarity values for samples in the pruned datasets are strictly smaller than 1.0 because samples with values greater than 1.0 are the ones that lie in the similarity gap and are pruned away.

Plotting the density of normalized similarities in Fig. 14 reveals that LAION-pruned has a much wider distribution with a smaller mode. Since normalized similarity closer to 1.0 indicates that the point lies closer to the similarity gap, a larger proportion of ImageNet-Train samples are close to the similarity gap compared to the proportion observed in LAION-200M.

We also compare the total number of points that are close to the similarity gap (normalized similarity > 0.9) in Tab. 6. Due to LAION's scale, each pruned LAION split contains 5-20 times more data points close to the similarity gap than ImageNet-Train. We expect that this large and diverse set of samples close to the boundary greatly dictates CLIP's performance.

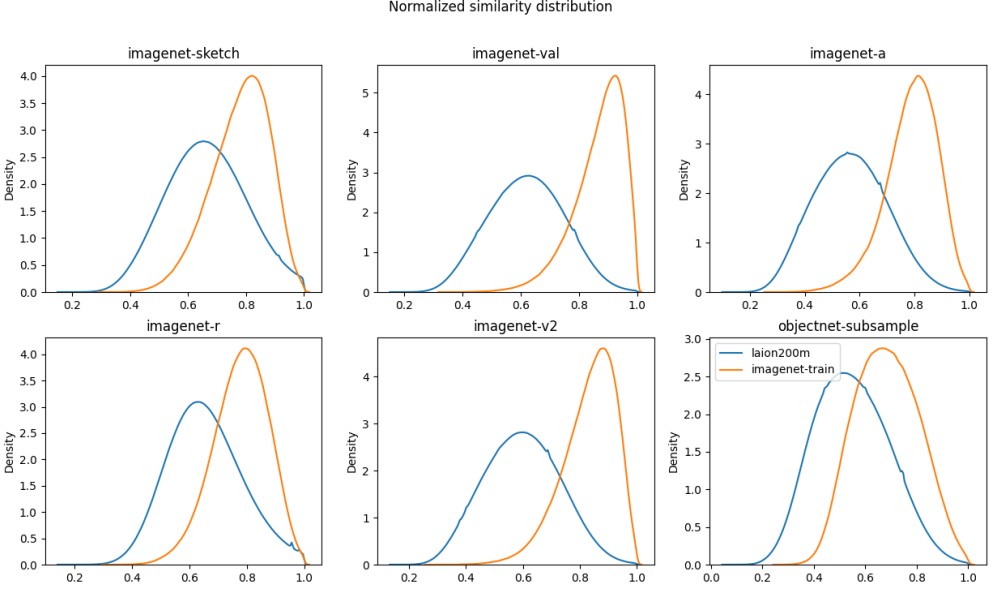

Figure 14: **Density of normalized similarity for LAION-pruned and ImageNet-Train**. We observe a wider density function for LAION-pruned with a smaller mode. This indicates that ImageNet-Train samples are generally more concentrated around the similarity gap.

Table 6: **Total number of samples in LAION-pruned splits and ImageNet that lie near the boundary of the similarity gap for each test set**. Closeness is defined by a normalized similarity > 0.9. While LAION-pruned samples are less concentrated around the gap (see Fig. 14), LAION-pruned still has 5 to 20 times more samples close to the boundary than ImageNet-Train.

| Dataset | LAION-200M | ImageNet-Train |
|---|---|---|
| ImageNet-Sketch | 8 859 133 | 131 087 |
| ImageNet-Val | 2 344 086 | 531 982 |
| ImageNet-A | 1 118 150 | 138 975 |
| ImageNet-R | 7 376 362 | 121 160 |
| ImageNet-V2 | 1 919 398 | 326 517 |
| ObjectNet | 1 558 301 | 52 277 |

# G  NEAREST NEIGHBOR VISUALIZATIONS

To generate nearest neighbors, we compute the nearest images of LAION in CLIP's (CLIP ViT-B/16+) image embedding space for each test image. After removing duplicates and near-duplicates within LAION, we visualize the top six images.

## G.1  LAION-400M VS IMAGENET-TRAIN

Just like in Fig. 1, we plot the nearest neighbors in LAION-400M and ImageNet-Train of random query images for each of the six datasets in Figs. 15, 16, 17, 18, 19, and 20.

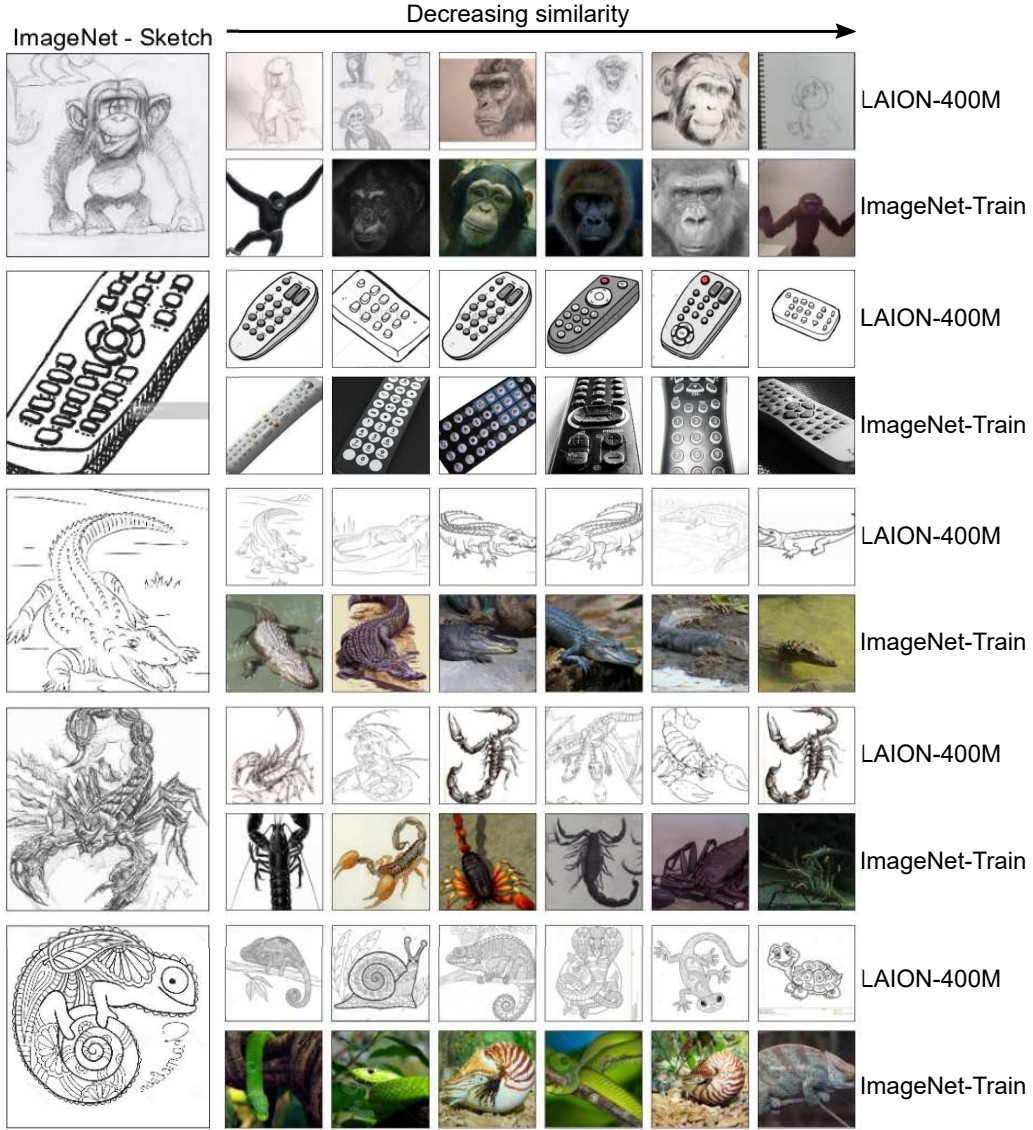

Figure 15: Nearest neighbors of *randomly* sampled ImageNet-Sketch queries in LAION-400M and ImageNet-Train ordered by decreasing perceptual similarity. We omit duplicates within the nearest neighbors. Perceptual similarity is computed in CLIP's image embedding space and can be considered to measure the "perceptual closeness" of images in terms of content and style.

## G.2 AFTER PRUNING

Tab. 7 reports the percentage of images in each of the six datasets that have higher similarity to LAION-200M/LAION-400M than ImageNet-Train. For each of the six test sets, we randomly sample query images that are more similar to LAION-200M than ImageNet-Train and plot the nearest neighbors in ImageNet-Train, LAION-200M, and LAION-200M after pruning by the respective test in Figures 21, 22, 23, 24, 25, and 26.

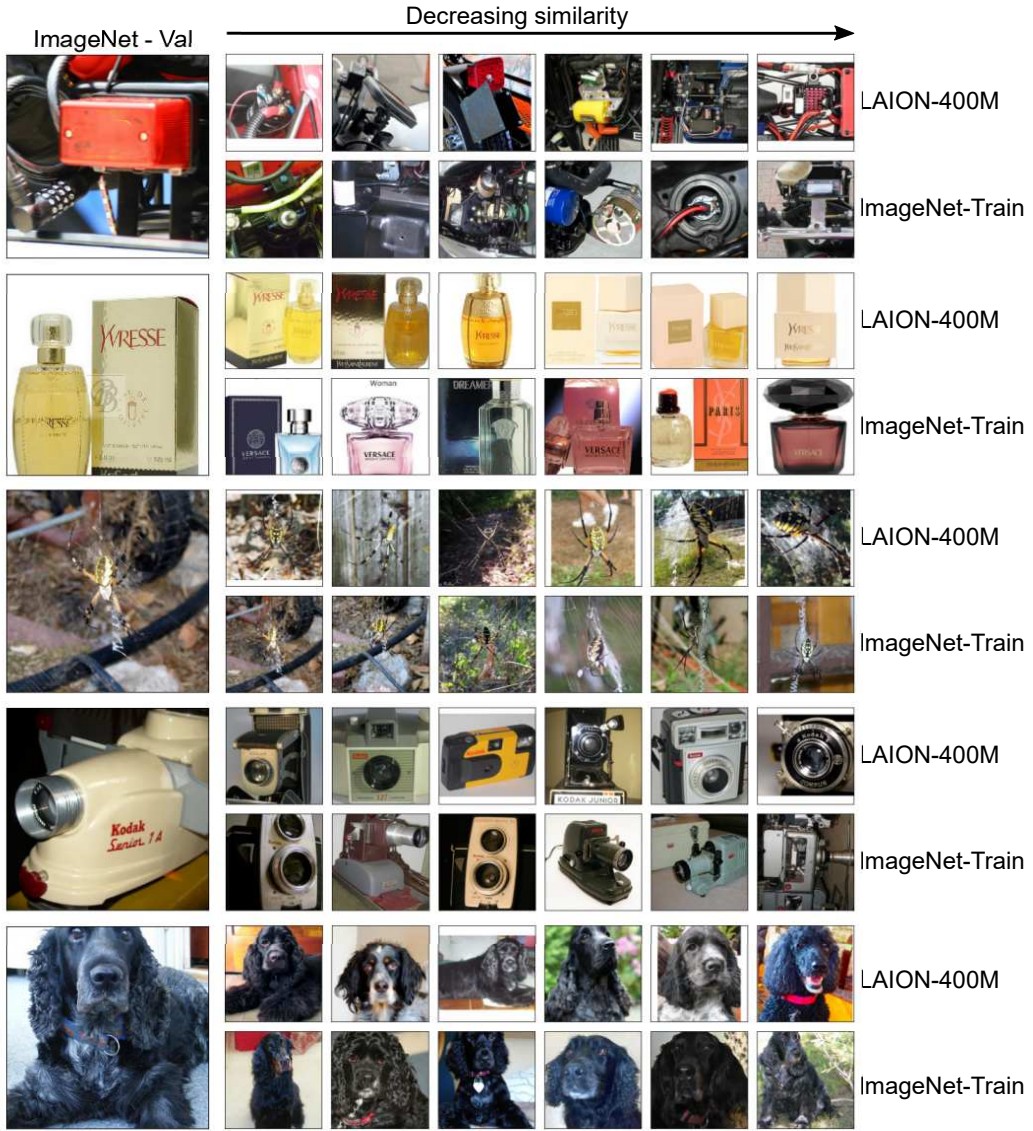

Figure 16: Nearest neighbors of *randomly* sampled ImageNet-Val queries in LAION-400M and ImageNet-Train ordered by decreasing perceptual similarity. We omit duplicates within the nearest neighbors. Perceptual similarity is computed in CLIP's image embedding space and can be considered to measure the "perceptual closeness" of images in terms of content and style.

Table 7: **Percentage (%) of points in the test datasets for which the nearest neighbor is in LAION-400M/LAION-200M rather than ImageNet-Train.**

| Dataset | Size | LAION-400M | LAION-200M |
| --- | --- | --- | --- |
| ImageNet-Val | 50 000 | 16.80 | 14.88 |
| ImageNet-Sketch | 50 889 | 97.94 | 97.45 |
| ImageNet-R | 30 000 | 87.88 | 86.74 |
| ImageNet-A | 7500 | 47.39 | 45.53 |
| ImageNet-V2 | 10 000 | 38.95 | 35.48 |
| ObjectNet | 18 574 | 63.24 | 61.62 |

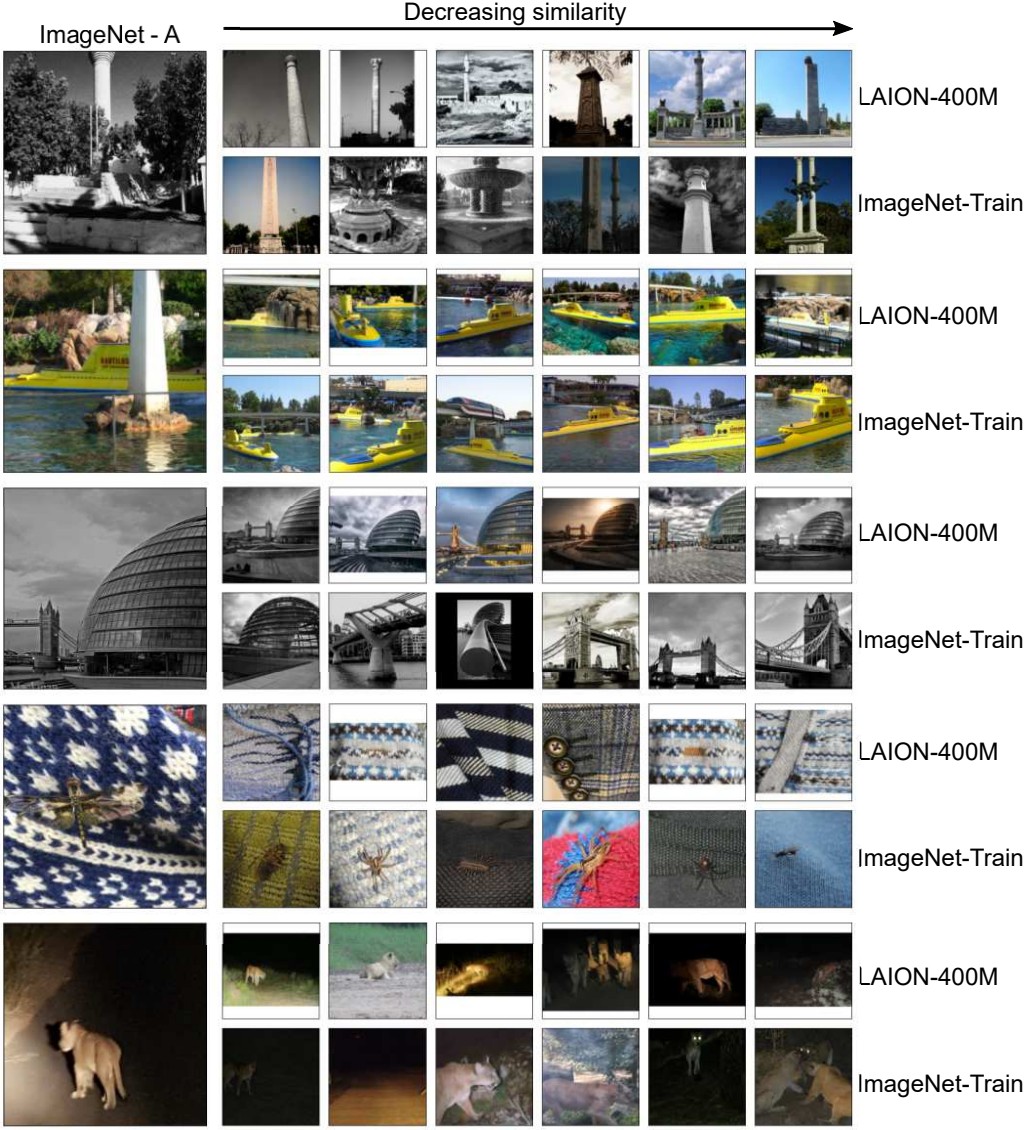

Figure 17: Nearest neighbors of *randomly* sampled ImageNet-A queries in LAION-400M and ImageNet-Train ordered by decreasing perceptual similarity. We omit duplicates within the nearest neighbors. Perceptual similarity is computed in CLIP's image embedding space and can be considered to measure the "perceptual closeness" of images in terms of content and style.

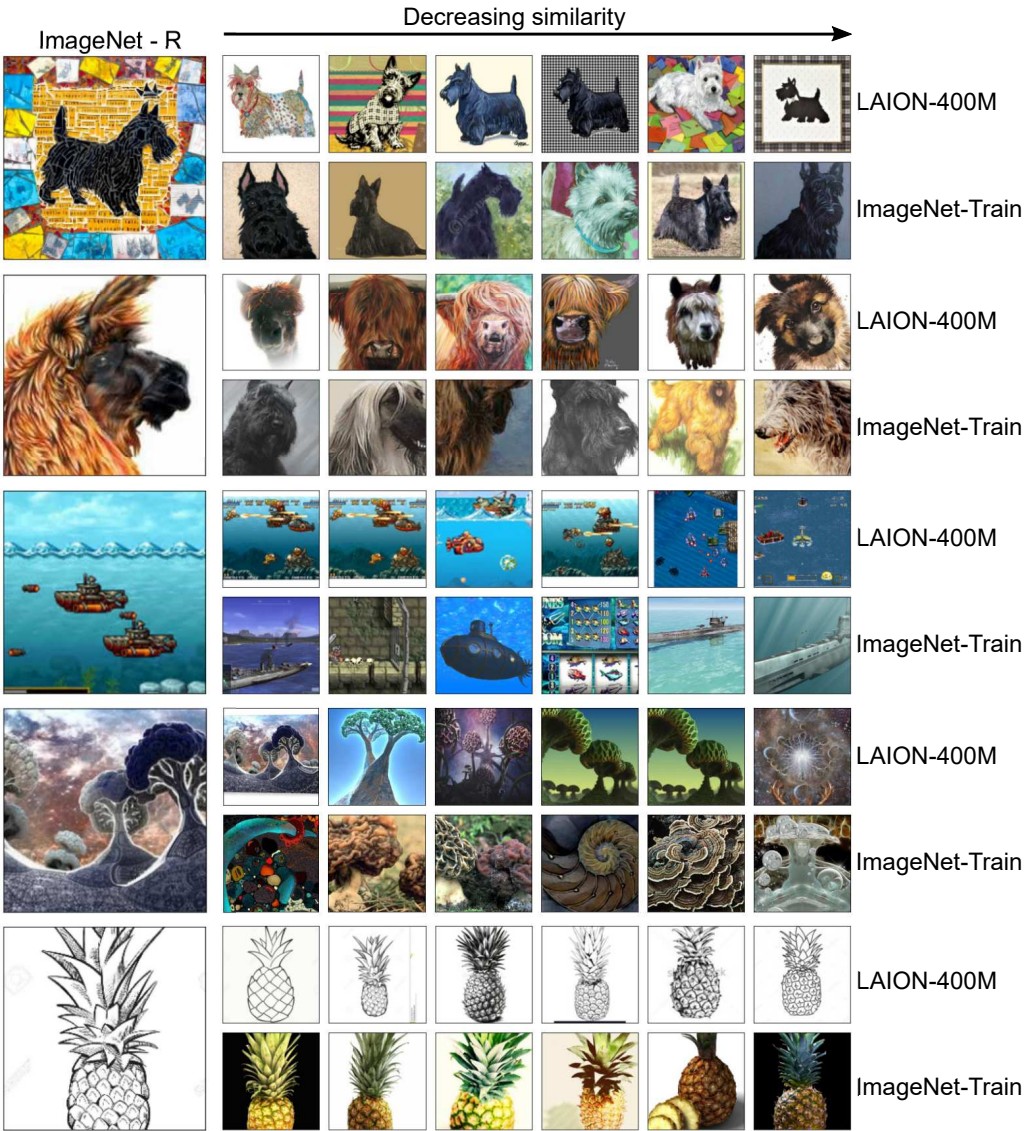

Figure 18: Nearest neighbors of *randomly* sampled ImageNet-R queries in LAION-400M and ImageNet-Train ordered by decreasing perceptual similarity. We omit duplicates within the nearest neighbors. Perceptual similarity is computed in CLIP's image embedding space and can be considered to measure the "perceptual closeness" of images in terms of content and style.

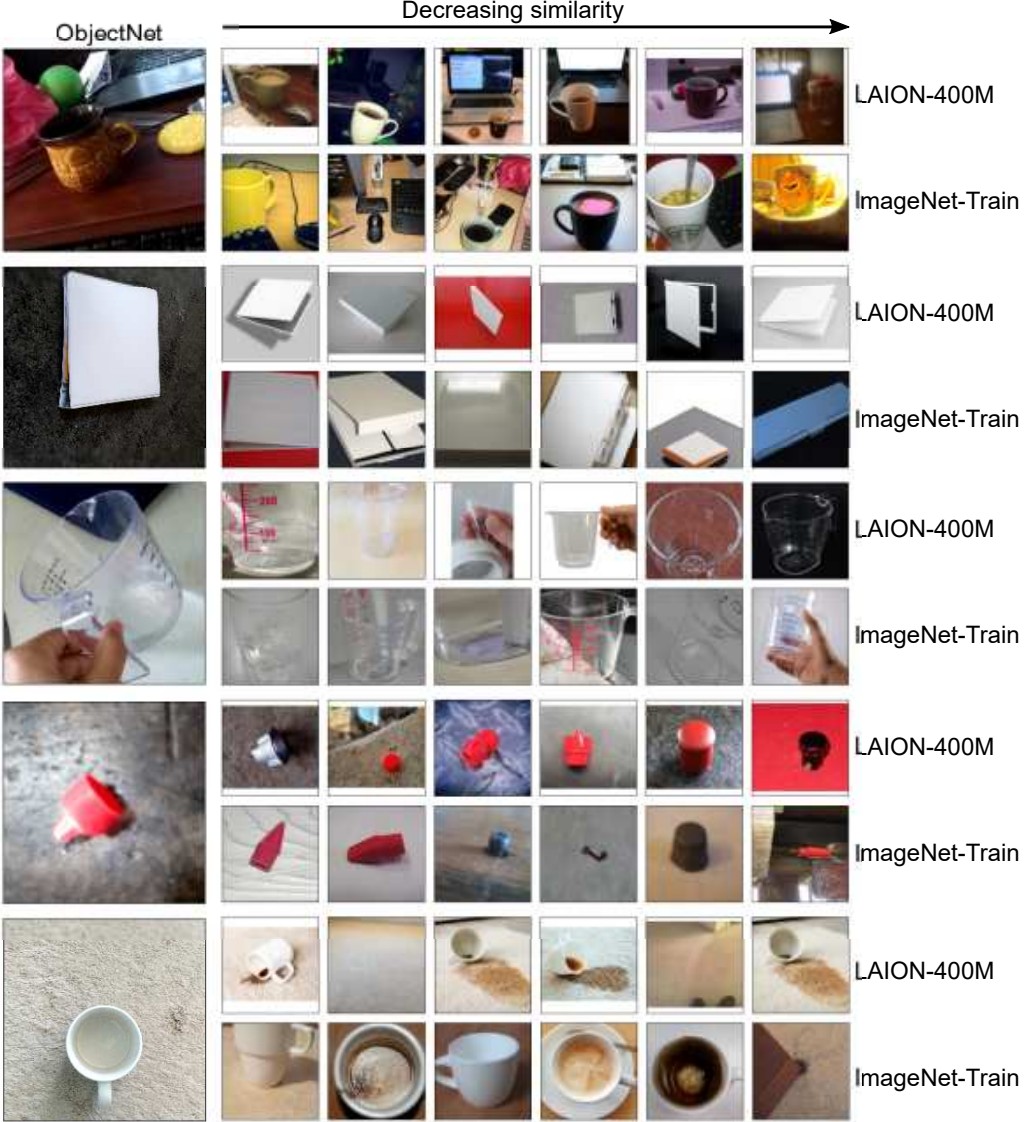

Figure 19: Nearest neighbors of *randomly* sampled ObjectNet queries in LAION-400M and ImageNet-Train ordered by decreasing perceptual similarity. We omit duplicates within the nearest neighbors. Perceptual similarity is computed in CLIP's image embedding space and can be considered to measure the "perceptual closeness" of images in terms of content and style.

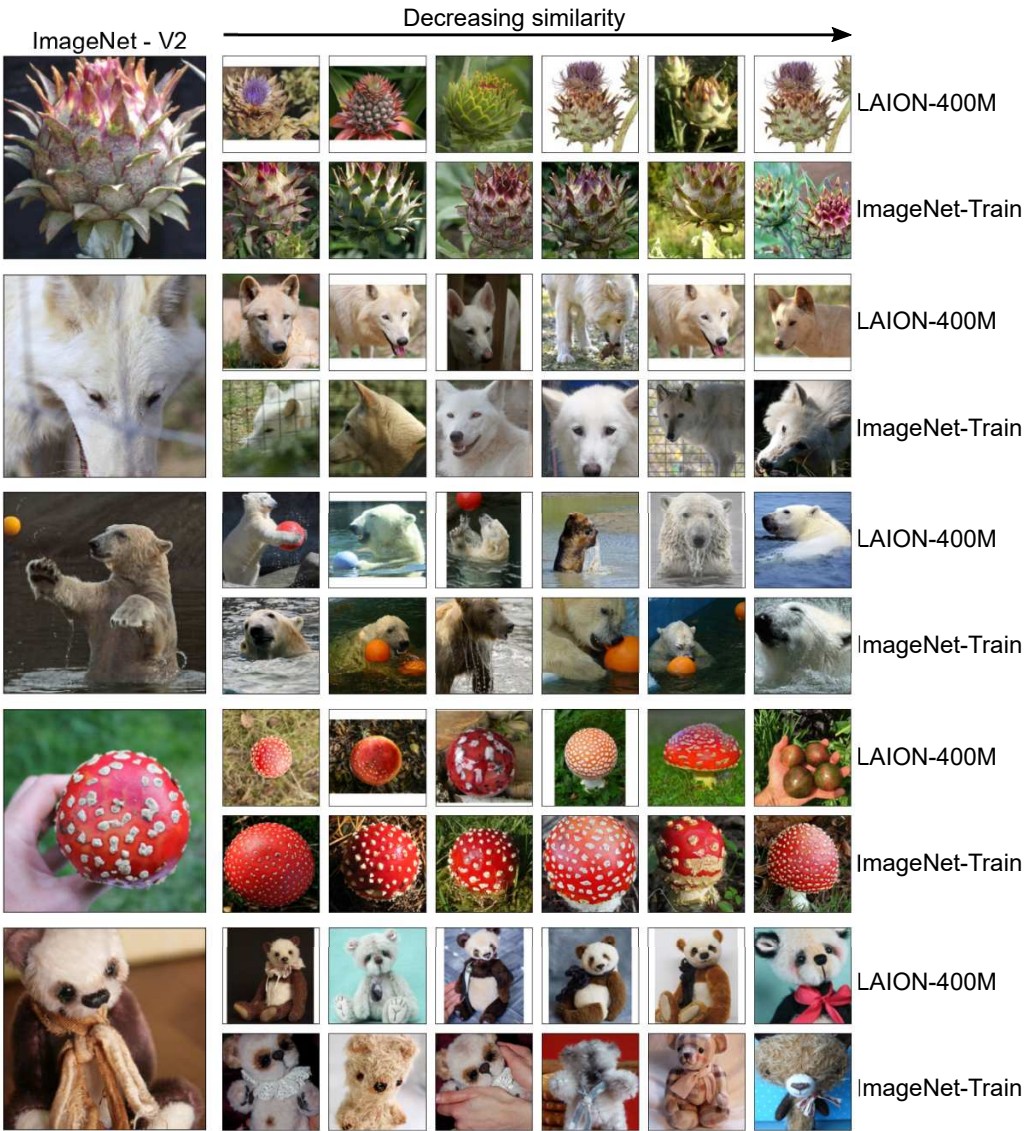

Figure 20: Nearest neighbors of *randomly* sampled ImageNet-V2 queries in LAION-400M and ImageNet-Train ordered by decreasing perceptual similarity. We omit duplicates within the nearest neighbors. Perceptual similarity is computed in CLIP's image embedding space and can be considered to measure the "perceptual closeness" of images in terms of content and style.

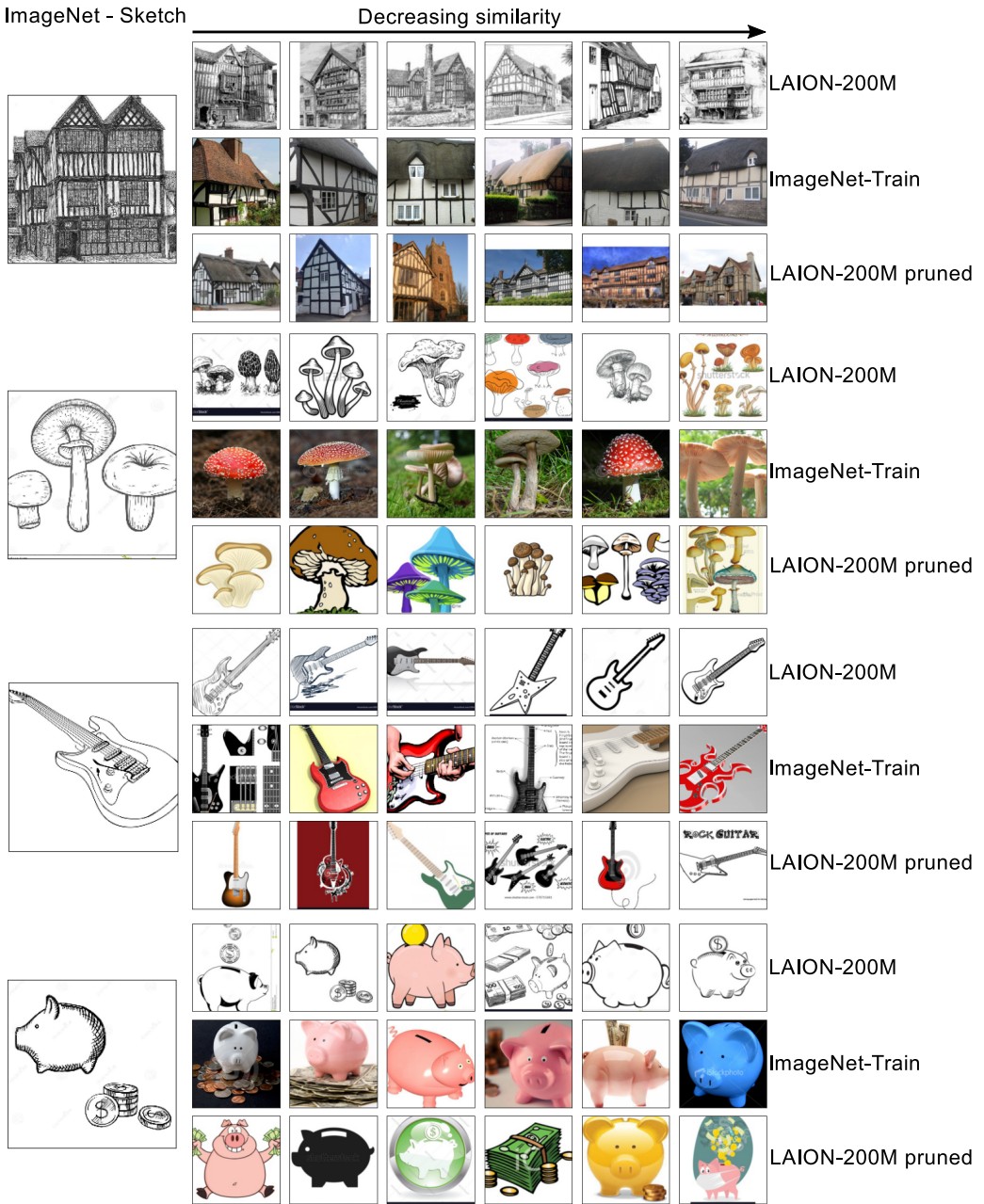

Figure 21: Nearest neighbors of ImageNet-Sketch images in LAION-200M, ImageNet-Train, and 'sketch-pruned' (LAION-200M pruned) ordered by decreasing perceptual similarity. The query (base) images are *randomly* sampled from the set of images that are more similar to LAION-200M than ImageNet-Train to see the effect of pruning (see Tab. 7). We omit duplicates within the nearest neighbors. Perceptual similarity is computed in CLIP's image embedding space and can be considered to measure the "perceptual closeness" of images in terms of content and style. LAION-200M clearly contains more similar images to samples in the test set compared to ImageNet-Train or 'sketch-pruned'.

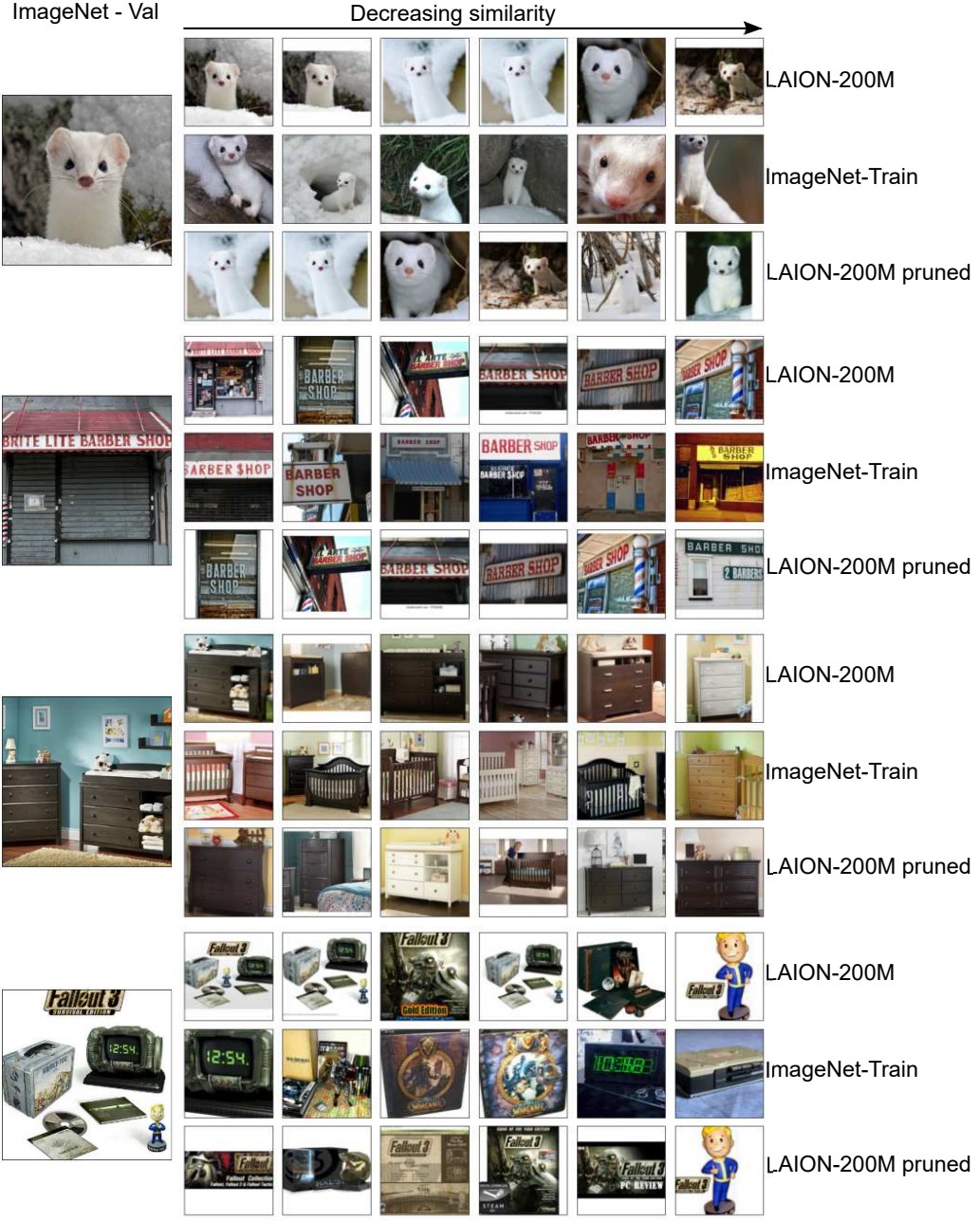

Figure 22: Nearest neighbors of ImageNet-Val images in LAION-200M, ImageNet-Train, and 'val-pruned' (LAION-200M pruned) ordered by decreasing perceptual similarity. The query (base) images are *randomly* sampled from the set of images that are more similar to LAION-200M than ImageNet-Train to see the effect of pruning (see Tab. 7). We omit duplicates within the nearest neighbors. Perceptual similarity is computed in CLIP's image embedding space and can be considered to measure the "perceptual closeness" of images in terms of content and style. LAION-200M clearly contains more similar images to samples in the test set compared to 'val-pruned'; ImageNet-Train images are in-distribution to ImageNet-Val and, therefore, contain similar samples.

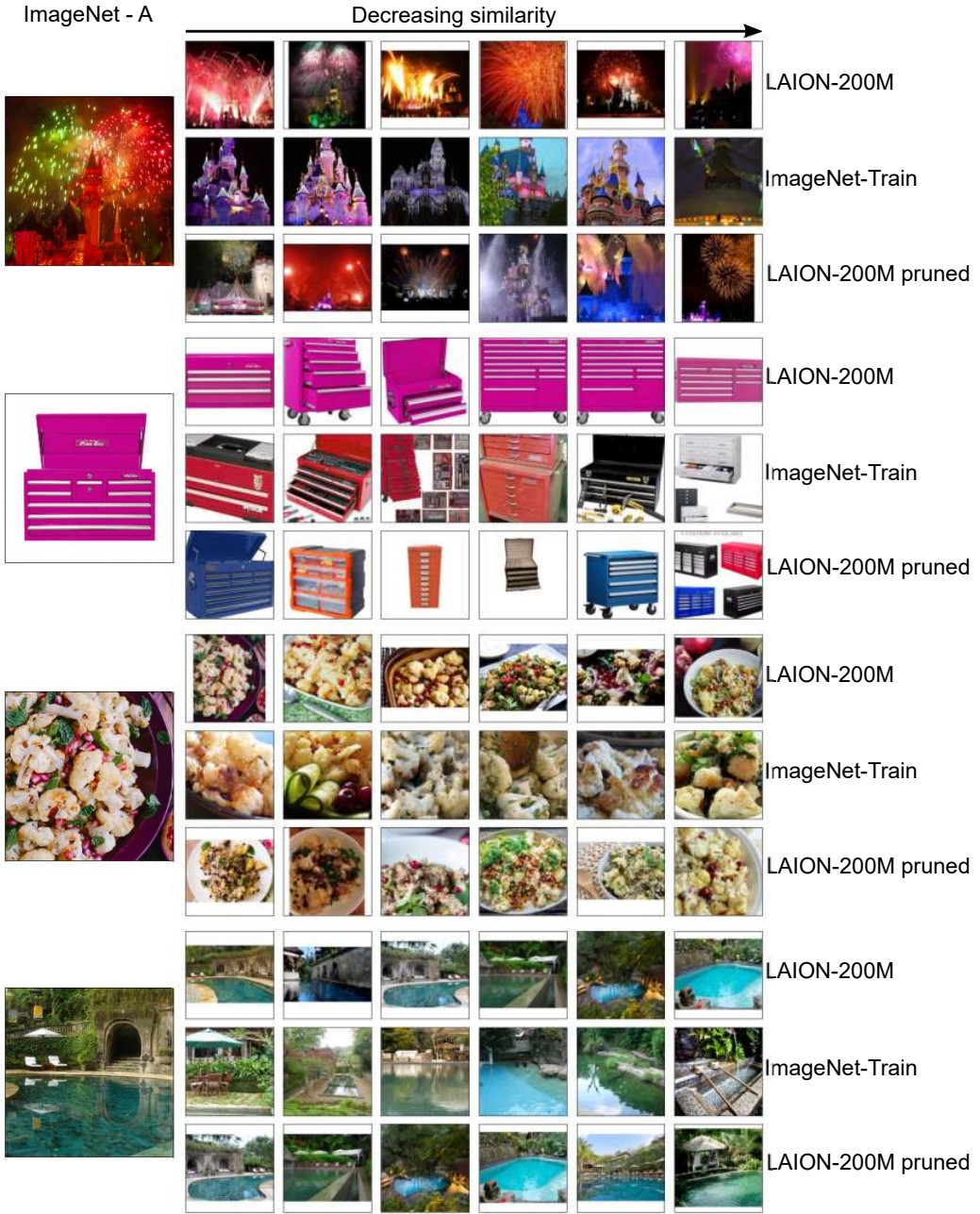

Figure 23: Nearest neighbors of ImageNet-A images in LAION-200M, ImageNet-Train, and 'a-pruned' (LAION-200M pruned) ordered by decreasing perceptual similarity. The query (base) images are *randomly* sampled from the set of images that are more similar to LAION-200M than ImageNet-Train to see the effect of pruning (see Tab. 7). We omit duplicates within the nearest neighbors. Perceptual similarity is computed in CLIP's image embedding space and can be considered to measure the "perceptual closeness" of images in terms of content and style. LAION-200M clearly contains more similar images to samples in the test set compared to 'val-pruned'; ImageNet-Train images are in-distribution to ImageNet-Val and, therefore, contain similar samples.

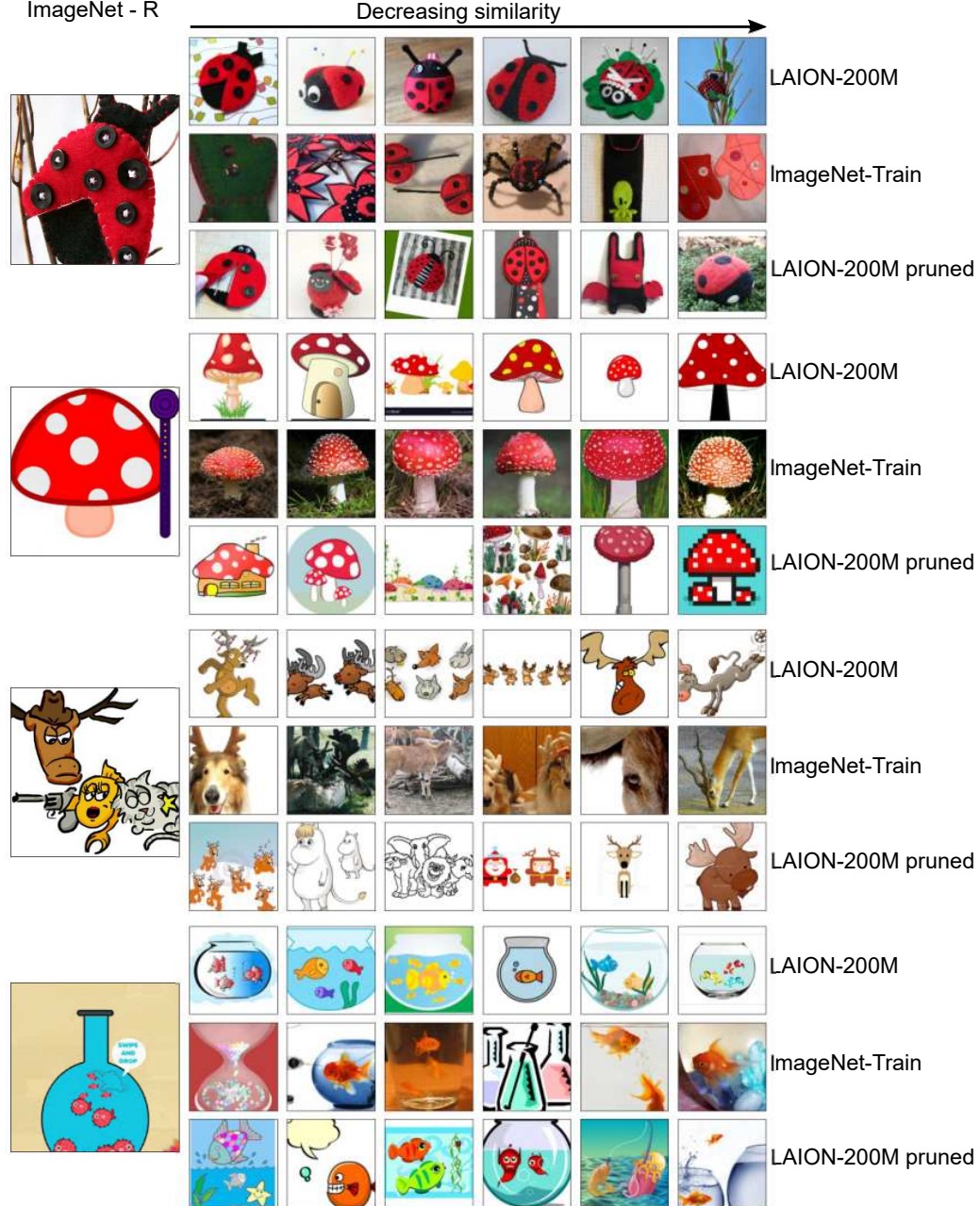

Figure 24: Nearest neighbors of ImageNet-R images in LAION-200M, ImageNet-Train, and 'r-pruned' (LAION-200M pruned) ordered by decreasing perceptual similarity. The query (base) images are *randomly* sampled from the set of images that are more similar to LAION-200M than ImageNet-Train to see the effect of pruning (see Tab. 7). We omit duplicates within the nearest neighbors. Perceptual similarity is computed in CLIP's image embedding space and can be considered to measure the "perceptual closeness" of images in terms of content and style. LAION-200M clearly contains more similar images to samples in the test set compared to 'val-pruned'; ImageNet-Train images are in-distribution to ImageNet-Val and, therefore, contain similar samples.

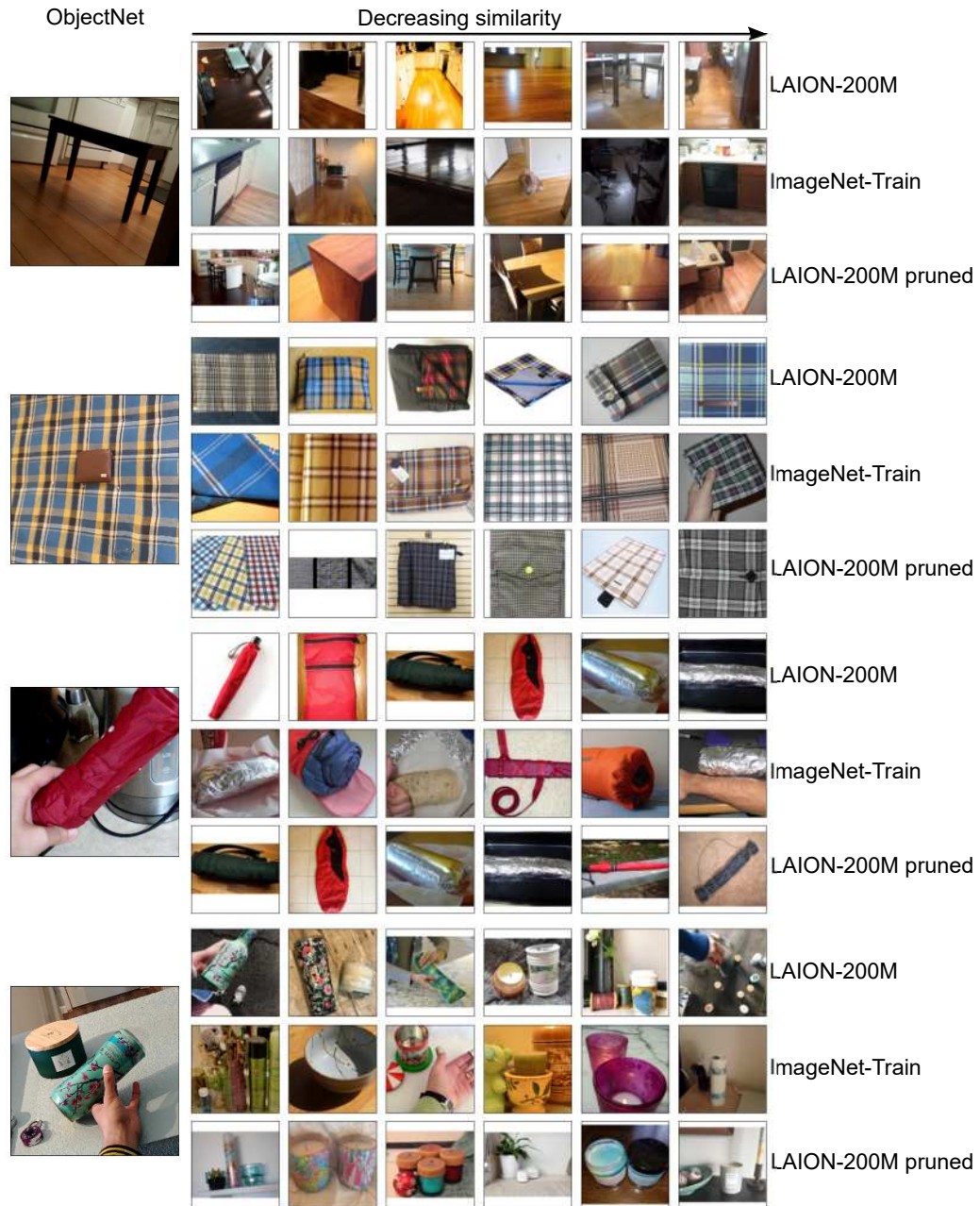

Figure 25: Nearest neighbors of ObjectNet images in LAION-200M, ImageNet-Train, and 'v2-pruned' (LAION-200M pruned) ordered by decreasing perceptual similarity. The query (base) images are *randomly* sampled from the set of images that are more similar to LAION-200M than ImageNet-Train to see the effect of pruning (see Tab. 7). We omit duplicates within the nearest neighbors. Perceptual similarity is computed in CLIP's image embedding space and can be considered to measure the "perceptual closeness" of images in terms of content and style. LAION-200M clearly contains more similar images to samples in the test set compared to 'val-pruned'; ImageNet-Train images are in-distribution to ImageNet-Val and, therefore, contain similar samples.

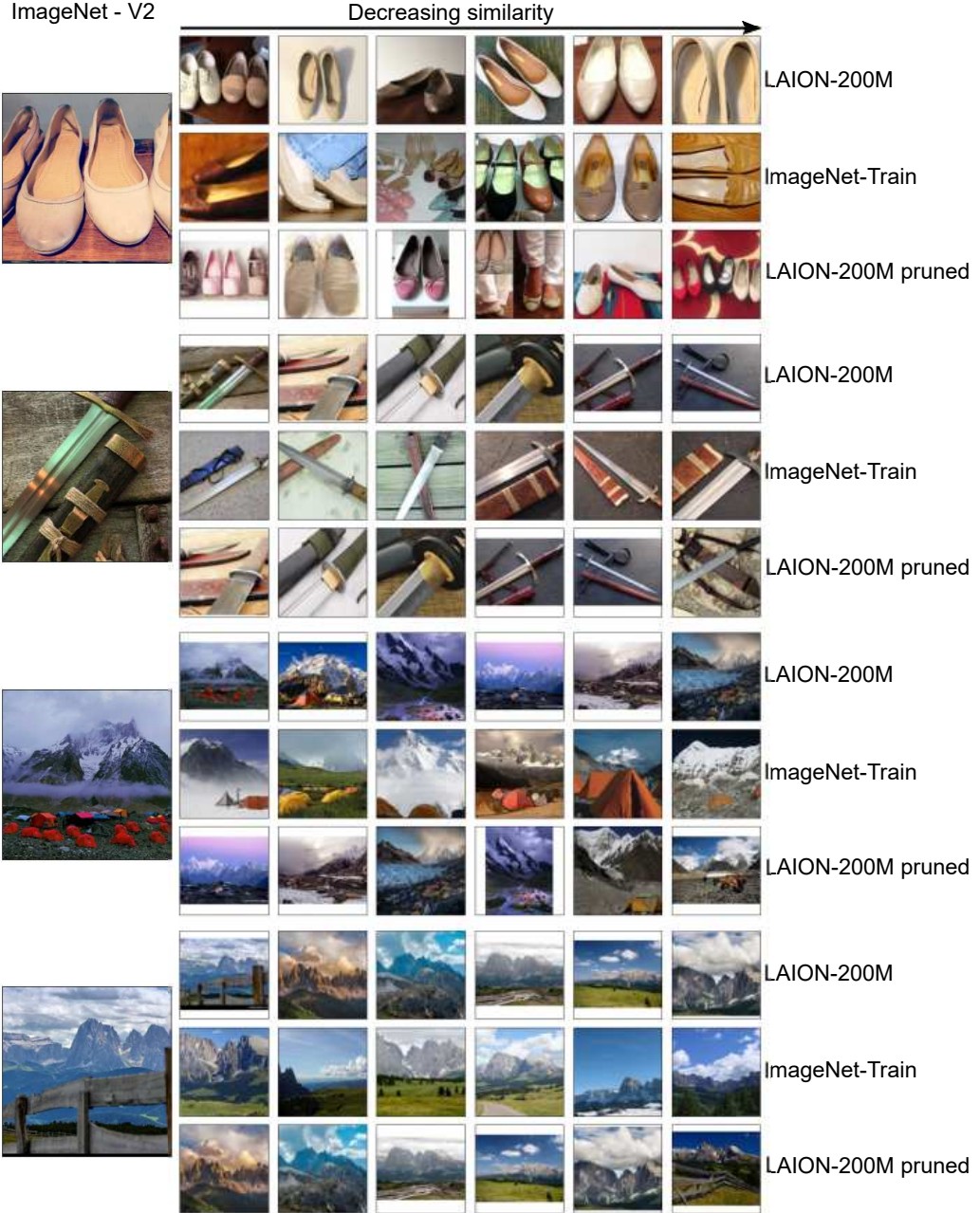

Figure 26: Nearest neighbors of ImageNet-V2 images in LAION-200M, ImageNet-Train, and 'objectnet-pruned' (LAION-200M pruned) ordered by decreasing perceptual similarity. The query (base) images are *randomly* sampled from the set of images that are more similar to LAION-200M than ImageNet-Train to see the effect of pruning (see Tab. 7). We omit duplicates within the nearest neighbors. Perceptual similarity is computed in CLIP's image embedding space and can be considered to measure the "perceptual closeness" of images in terms of content and style. LAION-200M clearly contains more similar images to samples in the test set compared to 'val-pruned'; ImageNet-Train images are in-distribution to ImageNet-Val and, therefore, contain similar samples.

