# OpenReview forum: "Does CLIP’s generalization performance mainly stem from high train-test similarity?"
_ICLR.cc/2024/Conference — ICLR 2024 poster_

### Official Review · Reviewer_X2Xy · 2023-10-30

**Soundness:** 3 good
**Presentation:** 3 good
**Contribution:** 2 fair
**Rating:** 5
**Confidence:** 5

**Summary:**

This paper studies the hypothesis “Does CLIP’s accuracy on test sets mainly stem from highly similar images in its train set?”. Their approach is to take a CLIP training dataset, LAION, and remove samples similar to OOD benchmarks, then retrain and evaluate the performance drop. In Section 4.1, they show that for some OOD datasets e.g., ImageNet-Sketch and ImageNet-R, there are more similar images to the OOD dataset in LAION than there are in the ImageNet-train set. These images are both semantically and stylistically more similar. They also observe a positive correlation between accuracy and having similar neighbors in the training set for each OOD benchmark. In the remainder of the paper, they prune the dataset by removing similar samples to OOD datasets and evaluate the performance of models trained on pruned datasets.

**Strengths:**

- Figure 1 clearly shows that the LAION-400M dataset contains semantically and stylistically similar images to OOD benchmarks while the ImageNet training set does not.
- Figure 3 quantitatively shows the nearest neighbors of OOD datasets in LAION-400M are on average more similar than the nearest neighbors from ImageNet-train set except for the Imagenet-Val set itself. Fig 3.right also clearly shows that CLIP model performs better on samples that it has seen similar images of it in the training set.
- Figure 4 shows that pruning LAION for similar samples to ImageNet-Sketch and ImageNet-Val results in substantial accuracy drop compared with pruning random samples. Hence showing that these similar samples are crucial for the effective robustness of CLIP models.

**Weaknesses:**

- Even though the results in Figure 4 show the near-pruned samples are crucial for OOD generalization of CLIP, they are not conclusive. In particular, the following two experiments would be useful:
i) Does a model *only* lose OOD generalization on one benchmark or are these near-pruned samples crucial for all sorts of image-classification performance on any benchmark? This requires plotting the accuracy on two datasets, e.g., ImageNet-Val and ImageNet-Sketch, and showing that after pruning samples similar to ImageNet-Sketch, we only lose performance on ImageNet-Sketch.
ii) Do CLIP models lose OOD generalization more quickly than classification models trained on ImageNet? For this, one would prune ImageNet with a similar procedure for X% of samples and compare whether the accuracy drop in ImageNet models is slower than CLIP models. If not, then we would know that any OOD generalization we have been seeing could be due to seeing similar samples in the training set.

- This paper is a good place to have a broader discussion on “What is zero-shot?” and “What is out-of-distribution robustness?” and I think the paper should expand on that. In particular, after showing that pruning loses OOD generalization, it is not clear what the community should do with these datasets and evaluation benchmarks. Should we say CLIP models have cheated and they are not zero-shot? Related to that, the objective of Sections 5 and 6 that “Correct for highly similar images” is not clear. Why would we want to prune our training datasets to get worse on some test benchmarks if those exact test samples do not appear in the training set? Why would we want models in Table 1 trained on pruned datasets that perform worse?

**Questions:**

- Figure 4: What if we perform a similar process for training on the ImageNet dataset? Do we observe a similar accuracy drop?
- Figure 7 shows that removing samples similar to ImageNet-Sketch and ImageNet-Val result in lower performance in all other ImageNet OOD datasets as well. Could this mean that these samples are important for a general understanding of the ImageNet distribution?
- Section 5: Would this method and any model trained on this dataset be considered as transductive learning? Because the test datasets would have been seen directly or indirectly by the model. Would this be against the license of datasets such as ObjectNet that say “ObjectNet may never be used to tune the parameters of any model.”?

Typos:
- Page 7: Let us for consider -> Let us consider

---

> ### Author Response · Authors · 2023-11-16
> **Part 1/2**
>
> We thank reviewer **X2Xy** for reviewing our paper and providing helpful suggestions. In light of your concerns, we would first like to clarify the scope of our work.
>
> It has been shown that data distribution drives CLIP’s performance on ImageNet OOD benchmarks [1]. The simplest hypothesis is that CLIP just performs well because LAION contains many **highly similar images** (i.e., images that are both semantically and stylistically similar) to the test datasets, possibly even direct duplicates. The main goal of our work is **to investigate if highly similar images drive CLIP to reach such a high performance on them**. Note that **to answer this hypothesis, we do not need to explicitly assume that the benchmarks are ID or OOD wrt. LAION.**
>
> ### Weakness 1 and Question 1: near-pruned samples are crucial for OOD generalization and Weakness 2, part 2: Objective of Secs. 5 and 6
>
> We firstly thank the reviewer for suggesting to do near/far pruning experiments on ImageNet. **We ran these experiments and now include them in App. B.2. (especially Figs. 8 and 9)**. Notably, based on Fig. 9, nearest neighbors seem to matter more for CLIP than for ImageNet as near/far pruning has a stronger absolute effect on CLIP. We hope these experiments address weakness 1.ii).
>
> Concerning weakness 1.i), we believe that Fig. 7 (in the appendix) already answers this: performance generally decreases across dataset. This is also true for the results of the main experiment summarized in Tab. 1.
>
> Yet, we maintain that we see our main contribution not as showing that nearest neighbors are crucial for OOD generalization, but rather that **highly similar images** cannot sufficiently explain CLIP’s performance on common benchmarks. In this context, the pruning in Sec. 4 merely demonstrates one important observation, and it is not our main contribution. We want to clarify the goals of Sec. 4:
>
> 1. We show that **similar images**, specifically the nearest neighbors, generally matter for CLIP’s performance (Figs. 3b and 4), whether the test set is ID or OOD. We don’t take ImageNet-Train into account here, and we don’t consider whether the pruned subsets are ID or OOD wrt. the test sets. This is emphasized by the new results in Figs. 8 and 9 where we can see that near-pruning ImageNet-Train with ImageNet-Val (which is ID wrt. ImageNet-Train) also hurts ResNet’s performance.
> 2. We want to highlight that LAION contains many **similar images**, but especially also contains **highly similar images**, i.e., images that resemble the test datapoints closer than any of the images in ImageNet-Train do (Figs. 1 and 3a).
>
> We make a distinction between **similar images** and **highly similar images**. In Sec. 4.2 and Fig. 4, we are pruning *any* images in the order of decreasing similarity, i.e., the pruning is unconstrained. In contrast, in the main experiment in Sec. 6 we only prune **highly similar images**, i.e. the pruning is constrained by the similarity thresholds from ImageNet. We formalize this distinction based on the generalization gap (since renamed to similarity gap) in Sec. 5.
>
> We hope that this clarifies the goal of our work and especially Secs. 5 and 6. We acknowledge that our introduction and some paragraphs throughout the paper were worded in a way that could give a wrong impression of our aim, and apologize for the confusion. **We have edited the manuscript in Secs. 1, 4, 5, and 6 to better reflect our intention**.

---

> > ### Author Response · Authors · 2023-11-16
> > **Part 2/2**
> >
> > ### Weakness 2: Zero-shot and OOD-robustness
> >
> > We understand the term ‘zero-shot’ as referring to transfer between tasks without additional fine-tuning. In the context of CLIP, this can indicate that the model is employed on a task (e.g., classification) it was not directly trained for. Often, this term is also used for transfers to a new dataset different from the pre-training dataset, but we don’t take the term to mean that the test set is OOD. Therefore, we don’t think our results should be taken to mean that CLIP cheated or to make a statement about its OOD generalization capability. We merely try to understand how certain datapoints in the training distribution affect performance in a classification task, specifically whether it is perceptual similarity of data points that drive this performance.
> >
> > In this context, we highlight again that we don’t propose our pruning method as a better training regimen or fairer way to train CLIP. Secs. 5 and 6 propose an intervention in the data selection process that allows us to study the impact of certain data points on classification performance in a controlled manner.
> >
> > ### Question 2: Are the pruned samples important for a general understanding of ImageNet?
> >
> > Fig. 7 shows that pruning highly similar samples to one dataset affects the performance on another dataset. This does indicate that samples similar to one dataset affect performance on other datasets. This is not surprising: Since the considered datasets are all based on ImageNet, they can be expected to share some characteristics. There are also some overlaps in the distribution shifts, e.g., ImageNet-R also contains drawings and sketches. Lastly, it should be noted that in these experiments, we prune without constraints and remove similar datapoints in LAION even beyond the generalization gap (now renamed to similarity gap) of ImageNet. When pruning tens of millions images, these similar images likely contain also plain natural images that resemble ImageNet-Train itself.
> >
> > ### Question 3: Can this be considered transductive learning? Does this violate the license of OOD datasets?
> >
> > We reiterate that the method proposed in our paper is simply a technique to analyze the effect of highly similar images on CLIP’s generalization performance, not a general method prescribed for training models. We believe that this does not constitute a breach of license. Moreover, we use the test datasets to prune away training points, not directly to update parameters. Therefore, we don’t believe our work to be transductive.
> >
> > ### Summary
> >
> > We hope to have addressed the reviewers concerns and to have cleared any confusion. We hope that the reviewer agrees that our work presents valuable findings that will be of interest to the community (especially considering that these experiments are difficult to run due to the time and compute needed to train CLIP) and can, therefore, confidently recommend accepting our paper.
> >
> > [1] https://arxiv.org/abs/2205.01397

---

> > > ### Comment · Reviewer_X2Xy · 2023-11-22
> > > **I thank authors for their response and their improvements. The results and writing still need improvements.**
> > >
> > > I acknowledge the improvements in the paper including the change of “generalization gap” to “similarity gap” and the authors’ clarification on the goal of Sections 5-6. However, the paper requires more improvements in i) writing to explain the goal, conclusions, and relations between results and ii) results to solidify the conclusions and understand the implications. As such, I retain my score.
> > >
> > > **W.1** The results in Figure 4 … are not conclusive. Two experiments would be useful:
> > > **W.1.i & Q.1** Does a model only lose OOD generalization on one benchmark…?
> > > Figure 7 shows the result for this experiment which is contrary to my expectation that near-pruning would result in accuracy drop only for one dataset. This raises a concern that the “similar images” that are removed are generally the most informative samples of the dataset as well. Maybe these samples are the highest quality samples and without them the CLIP model is not good according to any metric. To verify this, one can evaluate the performance of models in Figure 7 on tasks unrelated to ImageNet such as retrieval and other zero-shot classification tasks that are farther from ImageNet (e.g., other tasks in the open-clip/Datacomp eval suite with 38 tasks) . This is also related to the reviewer fHnd’s W.1 and reviewer fbpP’s Q.1 comment asking for additional experiments. I see that authors have provided new results on CelebA and Waterbirds, however, I’m asking for some other type of evaluation that does not require any new training but only evaluation on additional tasks.
> > >
> > > This point is also related to a concern about the validity of the metric used to define “similar images”. I see a relation to reviewer fbpP’s Q.2 and reviewer MLz9’s noted weaknesses. Specifically, I highlight that the distinction between “similar” and “highly similar” is not well-analyzed. For example, what are some examples that appear in both? What are some examples that appear in either one? Essentially, are “similar” images important for training and “highly similar” images are not.
> > >
> > > **W.1.ii** Do CLIP models lose OOD generalization more quickly than classification models trained on ImageNet?
> > > Figure 8 shows CLIP is more sensitive to near-pruning than is ResNet18 trained on ImageNet. I suggest replacing Figure 4 with Figure 8.
> > >
> > > **W.2.1** “What is zero-shot?”
> > > Given my new understanding of Section 5-6, I see that this question is not necessarily within the context of this work.
> > >
> > > **W.2.2 & Q.3** The objective of Sections 5 and 6 that “Correct for highly similar images” is not clear.
> > > The rebuttal response significantly resolved my confusion about these sections. I highly recommend including this response as part of the introduction. I read through the changes in the paper, but I still think someone reading the paper for the first time might be confused as to the objective of section 5 and 6 and what it is contributing after the observations of section 4. Importantly, the conclusions of Section 4 and 5-6 are somewhat in conflict with each other for which the authors make the distinction between “similar” and “highly similar” images. My understanding is that “similar images” are important to perform well on test sets but “highly similar images” are not. This also asks for more comparison between the “similar images” and “highly similar images”.
> > >
> > > **Q.2** Could this mean that these samples are important for a general understanding of the ImageNet distribution?
> > > > This is not surprising: Since the considered datasets are all based on ImageNet, they can be expected to share some characteristics.
> > > In fact I find this observation concerning given the conclusions of Section 5-6 and their contrast with Section 4 as discussed above. I suggest authors study questions I under in **W.1.i & Q.1** based on this observation.

---

> > > > ### Author Response · Authors · 2023-11-23
> > > > **Part 1/2**
> > > >
> > > > ### Restructuring and improving overall clarity
> > > >
> > > > We thank the reviewer for their detailed comments. Based on the reviewer’s suggestion, we have now improved the writing in the introduction and completely restructured the method part of our paper to improve clarity and better highlight how the individual experiments support our main conclusion.
> > > >
> > > > Specifically, we have integrated Sec. 5 into Sec. 4, and Sec. 4 is now structured as follows:
> > > >
> > > > - We first illustrate how perceptual similarity can be quantified in Sec. 4.1.
> > > > - Based on this metric, our data analysis and experimental results in Sec. 4.2 demonstrate that nearest-neighbor similarity in LAION-400M generally matter for CLIP’s performance.
> > > > - Sec. 4.3 shows how nearest-neighbor similarities differ between LAION-400M and ImageNet-Train, which leads to the hypothesis that this difference explains CLIP’s performance on ImageNet-based test sets. We note here that this similarity difference stems from *highly similar images*.
> > > > - Sec. 4.4 (previously Sec. 5) formalizes the notion of *highly similar images* in terms of the *similarity gap*, which leaves us with an interventional method to test our hypothesis.
> > > >
> > > > We opted not to substitute Fig. 4 with Fig. 8 because our primary message revolves around the significance of nearest neighbors in influencing CLIP's performance.
> > > >
> > > > We kindly suggest the reviewer to review the revised Secs 1, 4, and 5 and assess whether the core message of the paper is clearer now. Additionally, we identify what we believe to be the reviewer’s three main remaining concerns, and address them below.
> > > >
> > > > ### 1. Are the removed images generally the most informative samples for any test dataset?
> > > >
> > > > We thank the reviewer for suggesting this experiment and we agree with the reviewer that this analysis may strengthen the paper further. We added a new section B.2 to our Appendix, where we test the zero-shot classification accuracy on MNIST and SVHN of models trained on near/far pruned datasets with. Both MNIST and SVHN are sufficiently unlike ImageNet-like datasets. We acknowledge that further evaluation on other tasks (e.g. retrieval tasks) will strengthen this section and can implement those evaluations for the camera-ready.
> > > >
> > > > To re-state our previous results, in Fig. 7, we observe a consistent trend across all datasets that near-pruning with respect to either ImageNet-Sketch or ImageNet-Val decreases performance on ImageNet-Sketch and ImageNet-Val while performance on theses datasets is stable when doing far-pruning. In contrast, in Fig. 8, we observe that generally near-pruning LAION with ImageNet-Sketch/Val increases performance and far-pruning decreases it, respectively, on MNIST and SVHN. Thus, datapoints that are near-pruned **do** **not** comprise the “highest quality samples” and we don’t find evidence that “without them the CLIP model is not good according to any metric”. Pruning samples based on their similarity to ImageNet-Sketch or ImageNet-Val only decreases performance on ImageNet-like datasets.
> > > >
> > > > Note also that the observation that pruning decreases performance across benchmarks holds only for the unconstrained pruning of huge amounts of data in Fig. 3 (previously Fig. 4) and similar experiments. For the much more refined pruning in our main experiment, we do not generally see a performance drop across all metrics (compare Tab. 1).

---

> > > > > ### Author Response · Authors · 2023-11-23
> > > > > **Part 2/2**
> > > > >
> > > > > ### 2. How are similar and highly similar images different?
> > > > >
> > > > > Given the reviewer’s concerns, we have refined the characterization of **highly similar images** in the current version of the manuscript.
> > > > >
> > > > > While we previously also used the term **similar images**, we now remove this notion completely and throughout section four solely refer to the **nearest-neighbor similarity** of images, which we now properly define. Under this formalization, we can see that what was previously referred to as **similar images** is a relative concept lacking a strict characterization. Each training sample has a **nearest-neighbor similarity** attached to it, which is an entirely relative notion. For example, in our near-pruned experiments, we prune the top ‘N’ images based on the order of similarity to the test set. To elucidate further, for the ‘near-pruned-sketch’ 150M dataset, we prune the top 50M datapoints that are most similar (top 50M similarity values) to ImageNet-sketch. Likewise, ‘near-pruned-sketch’ 100M dataset, we prune the top 100M datapoints that are most similar to ImageNet-Sketch and so on.
> > > > >
> > > > > In contrast, **highly similar images** have a strict definition in the sense that they have higher similarity values to the test set than any ImageNet-train sample does. As a consequence, they share both semantics and style to the test dataset. Highly similar images are always pruned away for all datasets in Fig. 3 (previously Fig. 4), as we prune tens of millions of data points and **highly similar images** are top of the crop (the number of highly similar images that we prune for each test dataset is reported in Tab. 1). In the context of our work, especially Secs. 4.3 and 4.4 (previously Secs. 4.2 and 5) **highly similar images** are a proper subset of what we previously referred to as similar images.
> > > > >
> > > > > With this in mind, one can understand our main contribution in the following way. Of course, similarity of the training to the test set matters in general. This is not a novel insight and should be intuitive. Our initial experiments serve to underline this point and develop a tool kit that allows us to probe deeper into *how* similarity matters. A natural hypothesis is that it might simply be **highly similar images** that explain a bulk of the performance, as they share almost all characteristics of the test set. We test for this in a principled way and find this not to be the case.
> > > > >
> > > > > ### 3. Why does near-pruning with one test dataset impact model performance on another test dataset?
> > > > >
> > > > > As mentioned by the reviewer, it is reasonable to expect that near-pruning would result in an accuracy drop only for one dataset. **Highly similar images** are the ones that share both semantic and stylistic characteristics to the test datasets. When we are pruning by nearest-neighbor similarity, we obviously first prune the highly similar images, but as we keep pruning, the pruned images are less and less similar and we will prune only images with a few shared characteristics. For instance, when we do near-pruning of ImageNet-sketch of say 50M datapoints, the top of the crop would contain images very much alike ImageNet-sketch; then in the relatively lower order of similarity values, we would expect ImageNet-Val like images (i.e. images only with shared semantics or content but not style). Therefore, we see a performance drop on ImageNet-Val.
> > > > >
> > > > > However, when we only prune **highly similar images** (Sec. 5, previously Sec. 6), we see that the pruning is much more constrained and performance does indeed mostly deteriorate on the test set that was used for pruning (Tab. 1).
> > > > >
> > > > > We once again thank the reviewer for their time and feedback. We sincerely hope our response offers a clearer understanding of our work, potentially leading to a reassessment of its evaluation.

---

### Official Review · Reviewer_fHnd · 2023-10-31

**Soundness:** 3 good
**Presentation:** 4 excellent
**Contribution:** 3 good
**Rating:** 6
**Confidence:** 4

**Summary:**

This paper studies the generalization behaviour of CLIP from the perspective of the training set. CLIP, as a pioneering foundation model, is well known for its exceptional generalization capability. However, it remains unclear whether such a good generalization capability stems from the web-scale training set, as it may well enclose samples similar to those on the test sets. This paper tackles this question by filtering out similar samples according to CLIP-embedding-based nearest neighbors. By creating a LAION subset that comes with as large a generalization gap as the ImageNet-1K training set, this work concludes that it is other factors, rather than the model has already seen samples with similar distribution, that leads to the outstanding ood generalization of CLIP models.

**Strengths:**

1. This paper first studies the ood generalization behaviour of CLIP from the interesting perspective of train-test distribution similarity, and has drawn some intriguing conclusions, like a subset of 100M of LAION-400M is able to train a good-performing model on ood benchmarks.
2. The experiments are well motivated and designed, resulting in compelling results that even without nearest neighbor samples, CLIP is still to maintain good performance on ood benchmarks.

**Weaknesses:**

1. This paper only works on common classification ood benchmarks, but neglecting a whole bunch of other CLIP application domains. In the very least, retrieval is the most foundamental test ground to probe how good a CLIP model is. The authors could use MSCOCO as the in-domain dataset for sample filtering, and use Flickr-30k as the out-of-domain dataset. Also, experiments on more diverse benchmarks, like VTAB (check datacomp paper for current best practices), are encouraged.
2. Some crucial experimental details seem to be missing. For instance, how the nearest neighbor sets in section 4.1 are constructed are not mentioned at all (or mentioned in the appendix). How many samples are chosen in this set, and what is the threshold of CLIP score used here?
3. The authors should compare the similarity of nearest neighbors to test sets between the whole LAION datset (after filtering in section 5/6) and ImageNet-Train. Otherwise, one could argue that, even though the closest sample in LAION and ImageNet training set is about the same far away to the test set, LAION has a lot more samples close to the closest sample than ImageNet (but not as close to the test set as the closest sample), and thus still has a unfair advantage on those ood benchmarks, compromising the validity of the conclusion.

**Questions:**

What is the potential application of the finding in this paper? I know this is a pure analytical work, and believe its merit is above the aceeptance threshold of ICLR. But I am still intrigued to learn what is the broader impact of this work, like how this would enlighten future research or engineer endeavours.

---

> ### Author Response · Authors · 2023-11-17
> **Part 1/2**
>
> We are grateful to reviewer ****fHnd**** for reviewing our paper thoroughly and providing helpful suggestions. We address each of their concerns below.
>
> ### Weakness 1: On retrieval tasks and additional experiments
>
> We agree with the reviewer that one of the interesting use cases of CLIP is image retrieval. However, we do not believe that this undermines the relevance zero-shot classification. We focus on this task as there has been a huge body of work on this topic, and CLIP demonstrated unprecedented performance on ImageNet-based distribution shifts. Additionally, in contrast to classification, retrieval tasks are more complex and highly sensitive to captions, demanding an analysis that factors in both images and texts. Therefore, we did not perform experiments on Flickr30 and MSCOCO.
>
> We would also like to underline that not all ImageNet-based distribution shifts are characteristically similar. While ImageNet-Sketch and ImageNet-R have style changes, ObjectNet has shifts in rotation, background, viewpoints, and ImageNet-A has adversarial or hard images. Therefore our analysis and results are valid for a broad range of characteristically different distribution shifts in classification.
>
> That said, we agree with the reviewer that a stronger case can be made with experiments on other datasets and for distribution shifts that are not based on ImageNet. At the moment, **as an additional experiment in Appx. C, we have repeated the similarity analysis from Fig. 3 for CelebA [1] and Waterbirds [2]**. For CelebA, we split along the factors `Eyeglasses`, `Hats` as distribution shifts and zero-shot predict the factor `Male`. For Waterbirds, we can either split based on the background (`land` vs. `water`), or based on the combination of bird and land (`landbird-on-land/waterbird-on-water` vs. `landbird-on-water/waterbird-on-land`). We observe the same trends (similarity distributions to the train set/LAION differ, similarity is correlated with accuracy) as we did in Sec. 4.
>
> ```python
> | Dataset | Attributes | Correlation between similarity and zero-shot accuracy $\rho_S$ |
> |-|-|-|
> | CelebA | w/ eyeglasses | 1.0|
> | CelebA | w/ hat | 0.78|
> | | | |
> | Waterbirds | Coregroup | 0.82 |
> | Waterbirds| Land | 0.98 |
> ```
>
> Given the rebuttal’s tight time frame and the computational cost of training CLIP, we cannot yet replicate the experiment from Sec. 6 on these datasets. But, if the reviewer feels strongly about additional experiments, we are happy to repeat the complete analysis from Sec. 6 on CelebA and Waterbirds (and possibly also iWILDCam) for the camera-ready version.
>
> ### Weakness 2: Details for nearest neighbor visualization
>
> **We apologize for the confusion and have updated the manuscript to better explain what Sec. 4.1 is doing**.
>
> For the nearest neighbor sets in Fig. 1, we simply visualize the top six nearest neighbors to a given query sample (i.e., a sample from one of the test sets) in ImageNet-Train and LAION (after deduplication). **We have added more details in App. F regarding generating the nearest neighbors visualizations**.
>
> For the histograms in Fig 3., we simply take one of the six test sets and for each sample compute the similarity to its nearest neighbor in ImageNet or LAION (with duplicates removed). The histograms show the distribution over those similarities.
>
> Neither of these cases in Sec. 4.1. imposes a threshold on the perceptual similarity.

---

> > ### Author Response · Authors · 2023-11-17
> > **Part 2/2**
> >
> > ### Weakness 3: Similarity distribution of training datapoints before and after pruning:
> >
> > We thank the reviewer for suggesting to compare similarity distribution of the datasets after pruning, and we agree that this is an interesting analysis that strengthens our contribution. We have added the analysis in Appx. Section E.
> >
> > In this comparison, we show the nearest neighbor similarity of all samples in ImageNet/LAION as a ratio of the highest nearest neighbor similarity of the corresponding test sample. That is, points with a ratio  > 1 have been pruned, points with a ratio = 1 lie directly on the generalization gap (now renamed to similarity gap), and points with a ratio  < 1 lie farther away from the gap and remain after pruning. We observe:
> >
> > 1. Samples in ImageNet are proportionally much closer to the gap than samples in LAION.
> > 2. Due to its immense size, LAION nevertheless has more points close to the gap in absolute numbers.
> >
> > Thus, as the reviewer correctly notes, LAION’s large scale and data diversity give it **“an unfair advantage on the OOD benchmarks”**. However, we reiterate that this is not in conflict with our findings and instead corroborates our conclusion: **We aimed to investigate to what extent *highly similar training points (images that are both stylistically and semantically very close to the test points)* factor into CLIP’s performance**. We do not claim that our work examines all properties of the data distribution; in fact, we already hypothesize in the conclusion section that after correcting for this factor **data diversity** and **data scale** are the most likely remaining confounders for a fair comparison. Given the complexity and computation cost of our experiments, we leave further analysis of these confounders for future work.
> >
> > ### Question 1: Potential application and impact
> >
> > We see our work as one step towards the bigger question **“How do we build pretraining datasets that yield good downstream performance?”**. The role of **highly similar images** is of course just one piece of the puzzle. We believe it to be a good starting point as the simple hypothesis that “CLIP has just seen everything” is well-defined and testable, but (to our knowledge) has not been addressed yet.
> >
> > Given the size of the datasets and models involved, we believe that any insight into the design of pretraining datasets is useful. As also stated in the paper, **our analysis indicates that data diversity may be more important than highly similar images for generalization** and so could be interesting to examine next. Our released coreset hopefully provides researchers with a way to study this and related hypotheses more easily.
> >
> > Ultimately, **we hope to understand the interplay between training data and downstream performance** well enough to be able to guide the curation of new datasets that can be smaller and more effective.
> >
> > ### Summary
> >
> > We hope that our detailed explanation and the supplementary experiments have adequately addressed the questions and concerns raised. We sincerely hope this additional information offers a clearer understanding of our work, potentially leading to a reassessment of its evaluation.
> >
> > [1] https://mmlab.ie.cuhk.edu.hk/projects/CelebA.html
> >
> > [2] https://github.com/kohpangwei/group_DRO

---

> > > ### Comment · Reviewer_fHnd · 2023-11-19
> > >
> > > Thanks for the prompt reply!
> > >
> > > (1) I recognize the high computational cost of CLIP may restrict the authors from repeating experiments on other datasets.  But I still feel the analysis on retrieval could greatly enhance the applicability of this work. For instance, a normal retrievel model trained on MSCOCO may fail to generalize to Flickr30K, but CLIP can. Even if caption results in another dimension of complexity, probing on what property of LAION of CLIP leads to that generalizability would be highly vauable. Plus, it is always possible to use a super strong text encoder that handles all captions in MSCOCO and Flickr30k very well. The authors are encouraged to add more experiments, or at leat include a related discussion in the camera-ready revision.
> > >
> > > (2) I agree with the authors that even after pruning, the LAION dataset may still contain more samples similar to the test set due to its enormous scale and diversity. This somewhat is aligned with the commment from Reviewer MLz9 that even though the samples with the exact attributes are removed, composite attributes might still be informative enough for the model to learn related concept. The authors are encouraged to include a related discussion in the camera-ready revision.
> > >
> > > My original rating still applies.

---

> > > > ### Author Response · Authors · 2023-11-21
> > > >
> > > > We greatly appreciate your reply and comments. As per your suggestion, we will conduct additional experiments on retrieval task (MSCOCO/Flickr-30k) and add a related section for the camera-ready version.

---

### Official Review · Reviewer_MLz9 · 2023-10-31

**Soundness:** 2 fair
**Presentation:** 2 fair
**Contribution:** 3 good
**Rating:** 6
**Confidence:** 4

**Summary:**

The goal of this work is to understand whether the superior OOD performance of foundation models like CLIP is a result of the training dataset containing images that are very similar to the OOD test set. Towards this, the authors systematically create several splits of the base LAION dataset that was used for training OpenCLIP. Firstly, they find that pruning samples that are very similar to the OOD test sets results in a considerable drop in the OOD performance. However, by matching the train-test similarity with that of ImageNet for a fair comparison, the authors find that CLIP still shows significant gains when compared to an ImageNet pretrained model. Thus, although LAION contains images that are very similar to ImageNet-OOD test sets, this is not the key reason for better OOD generalization of CLIP. Understanding the reasons for better generalization of CLIP still remains an open question.

**Strengths:**

- The key finding that despite reducing the similarity between the training data and the test sets, there is an improvement in test set performance, is helpful.
- Several insightful experimental results are presented, which is helpful for the community, especially given that these experiments are very computationally intensive.
- The pruned datasets whose code is released, can help with further investigation on why CLIP models have better OOD performance.

**Weaknesses:**

- Although the results are interesting, my main concern is that the analysis is not sufficient to enable fair **OOD** testing. If the paper was about **ID** performance alone, removing train set samples that are similar to each test sample would have been sufficient, as reported in the paper. However, **OOD** implies that the **distribution** of images is unknown. So, to actually conclude that OOD evaluation is fair, all images from the test **domain** should have been removed, not only the images that are similar to every test set image. Therefore, as mentioned in the abstract, the term "out-of-distribution generalization" is still not meaningful even by training on the pruned datasets considered in the paper.
- To elaborate, if there are 5 sketches of the class "airplane" in the dataset, all images that are close to these sketches are removed. But there may be other airplane sketches, which are farther away than the closest image in ImageNet-test set, which are not removed. Thus, it is not guaranteed that all images of the given classes and test **domain** are removed from the train set.
- Further, there may be other objects, such as "space shuttle"  which are not included in ImageNet, thus sketches of space shuttles can be present in the training data, in addition to natural images of the same. Thus the domain "sketch" is not unseen by CLIP, and hence the evaluation is not truly OOD.

Minor feedback:
- Clarity of the abstract and contributions list needs improvement. It would be better to make a shorter summary of the key contributions.
- It would be better to give a different name to "generalization gap" as this term is used in a different context.

Typo:

"We provide anecdotal evidence in Fig. 1 where we choose samples from ImageNet-Sketch and ImageNet-R and examine their nearest perceptual neighbors in LAION-400M and **ImageNet**"

**Questions:**

Several additional experiments are required to be done, in order to emulate a true **OOD** setting, as discussed in the weaknesses section.
For example, one should remove all "sketch" style images as well, from L-200M+IN-Train (sketch-pruned) in order to test on the ImageNet-sketch test set for a true OOD evaluation. Otherwise, other sketch images that are present in the dataset can help bridge the domain gap between sketches and natural images, leading to an unfair OOD evaluation.

---

> ### Author Response · Authors · 2023-11-16
>
> We thank reviewer **MLz9** for reviewing our paper, calling our experiments insightful, and providing helpful suggestions. In light of your concerns, we would first like to clarify the scope of our work.
>
> Prior work has established that data distribution drives CLIP’s performance on ImageNet Out-of Distribution (OOD) benchmarks [1]. While precisely assessing how similar image distributions are is difficult, and (to the best of our knowledge), there is no consensus how the OOD-ness of a dataset could be quantified, it seems evident that these ImageNet OOD benchmarks cannot be considered to be OOD wrt. LAION. But **independent of whether these benchmarks can be considered OOD or not, it is interesting to examine what properties of its training distribution allow CLIP to reach such a high performance on them**.
>
> The simplest hypothesis (a first-order explanation, if you will) is that CLIP just performs well because LAION contains many **highly similar images**, i.e images that are both semantically and stylistically similar to the test datasets, possibly even direct duplicates. Note that **to answer this hypothesis, we do not need to explicitly assume that the benchmarks are ID or OOD wrt. LAION**, neither before nor after our pruning intervention.
>
> To summarize: **We agree with the reviewer that measuring CLIP’s “true OOD performance” is  interesting, but we do not claim to address this question in our work.** We realize that our introduction and some paragraphs throughout the paper were worded in a way that could give this impression, and apologize for the confusion. **We have edited the manuscript in Secs. 1, 4, 5, and 6 to better reflect our intention**. We would now like to also address the individual concerns here.
>
>
> ### Weaknesses 1, 2, 3 & Question 1: On a fair OOD generalization setting by removing all images of a certain domain
>
> We completely agree with the reviewer that fair OOD testing would require removing all images pertaining to a certain domain. Indeed, we initially considered doing exactly this, but had to realize that this is far from trivial.
>
> To remove all images of a certain domain, we need to be able to label each image as ‘ID’ or ‘OOD’. This essentially means that we need access to a domain classifier (which would also need to have near-perfect accuracy so that no images are overlooked). However, it is very unclear how such a classifier could be obtained. Even for the ‘sketch’ domain, where a classifier could conceivably be trained, it is unclear exactly how the classifier should demarcate this domain: Should the domain contain all sketches, even sketches with characteristics not present in ImageNet-Sketch? What about pencil sketches with a very limited color-palette (ImageNet-Sketch is mostly black-and-white)? What about tattoos or small sketches on objects in natural images (like printing on t-shirts? For other benchmarks, such as ImageNet-A, it is even less clear how the test images constitute a well-separable domain of images. It is precisely this vagueness in defining a domain based on a given test set that prevents us from building a fair OOD setting, which is why we do not claim to analyze this.
>
> The reviewers’ example also touches on another interesting point: It could well be that after our pruning sketches of, say, space shuttles remain, while airplanes are only present in natural images. In this case, CLIP might have to compositionally generalize in order to reach high performance on the ‘airplane’ class in ImageNet-Sketch. It would indeed be very interesting to analyze how individual concepts in the data distribution can be combined by the model, but this is out of the scope of this work, and we are not even sure that such an analysis would be computationally feasible.
>
>
> ### Weakness 4: Clarity of the abstract and contributions list needs improvement
>
> We have improved the abstract and the contributions section.
>
>
> ### Weakness 5:  It would be better to give a different name to "generalization gap" as this term is used in a different context
>
> We have changed “generalization gap” to “similarity gap”.
>
>
> ### Summary
>
> Given this clarification, and considering that the reviewer
> - evaluates our “key finding” as “helpful”,
> - regards our “experimental results” as “insightful” and “helpful for the community, especially given that these experiments are very computationally intensive”,
> - and agrees that the released code can “help with further investigations”,
>
> we ask whether the reviewer might reconsider their evaluation to make our results accessible to the community.
>
> [1] https://arxiv.org/abs/2205.01397

---

> > ### Comment · Reviewer_MLz9 · 2023-11-16
> >
> > I thank the authors for their response and the changes in the draft. I agree that a true OOD evaluation of CLIP has several challenges as discussed in the response. I encourage the authors to include a detailed discussion on this in a limitations section. This would be helpful for future research towards developing methods to evaluate the true OOD robustness of CLIP, and also to build a set of rules on training and test datasets to evaluate OOD robustness of such models. I update the rating to 6 based on the response.

---

> > > ### Author Response · Authors · 2023-11-21
> > >
> > > We greatly appreciate your help in improving our manuscript and sharpening our core message. We also agree that a discussion on what constitutes an OOD domain is interesting and important for future work. Therefore, we have included a paragraph discussing this question in our Discussion section.

---

### Official Review · Reviewer_fbpP · 2023-10-31

**Soundness:** 3 good
**Presentation:** 3 good
**Contribution:** 3 good
**Rating:** 6
**Confidence:** 4

**Summary:**

This paper tries to uncover the reason for the good generalization of foundation models like CLIP. Specifically, we see that CLIP has a good zero-shot accuracy on OOD datasets, which is much higher compared to the models that are first trained on the labeled dataset (ID), such as ImageNet, and then tested on OOD datasets, such as ImageNet-Sketch.

As the pre-training data is large, it is highly likely to contain similar images to the OOD dataset. If that is the case, we might not call the test OOD dataset to be truly out-of-distribution, which might be the reason for the good performance of CLIP.

Authors try to find an answer to the above problem through various experiments.

They define the perceptual similarity of two images using the CLIP ViT-B/16 embedding of a pre-trained CLIP model. They find that the general OOD datasets like ImageNet Sketch have more common images with the LAION dataset than the ImageNet dataset.

In the first experiment, the authors removed images similar to the OOD dataset from the LAION dataset iteratively and measured the performance of models trained on a pruned dataset. As expected, they found a general trend of performance decrease, indicating that similar images in the training dataset had more importance.

The second experiment compared the models trained on small datasets such as ImageNet vs those trained on LAION. The authors define a metric called generalization gap, which is the set of minimum distance of each image in the test set with every image in the training set. They then remove the data points from the larger dataset (here, LAION) to have the same generalization gap as that of the ImageNet with the OOD dataset. They then trained a CLIP model on the pruned dataset.
They discovered some performance drops, but the performance was still high compared to the case where the model was just trained on the ImageNet training set. This led to them concluding that the high performance of the CLIP is mostly due to more training data rather than test train similarity.

**Strengths:**

The paper is very well written. The experiments are clearly defined, and the motivation for each experiment is mentioned.

**Weaknesses:**

Even though authors perform many experiments, the experiments are performed on a single type of dataset, which are the variations of the ImageNet.
It would have been to include experiments with other OOD datasets such as iWILDCam or FMoW. Without these experiments it is not sure if these analysis is limited to one type of dataset.

**Questions:**

1. Are there experiments on other datasets unrelated to ImageNet? Having these experiments would be better to make a strong case.
2. Why do we use only CLIP ViT-B/16 embeddings to filter? Won't larger CLIP models have a better embedding to filter out the data points?

---

> ### Author Response · Authors · 2023-11-17
>
> We express our gratitude to reviewer **fbpP** for thoroughly reviewing our paper, lauding our paper’s clarity, and providing helpful suggestions. We would like to address each of your individual concerns here.
>
> ### Weakness 1 & Question 1: Experiments on other datasets which are less ImageNet-like
>
> We thank the reviewer for the suggestion to perform experiments on iWILDCam or FMoW. **We agree that experiments on distribution shifts not based on ImageNet are interesting to look at.** To provide additional context: We chose ImageNet-based distribution shifts because CLIP models perform extremely well on those, making them perfect candidates to test our primary hypothesis: whether **highly similar images drive performance**. We would also like to underline that not all ImageNet based distribution shifts are characteristically similar. While ImageNet-Sketch and ImageNet-R have style changes, ObjectNet has shifts in rotation/background/viewpoints, and ImageNet-A has adversarial or hard images. Therefore, our analysis and results are valid for a broad range of characteristically different distribution shifts.
>
> That said, we agree with the reviewer that a stronger case can be made with experiments on other datasets. We find the proposed iWILDCam and FMoW datasets problematic since CLIP’s  (ViT-B/32) zero-shot performance on them is rather low (7.45% and 12.96%) [2]. As an alternative, we propose to look at two datasets with higher CLIP zero-shot accuracy. On CelebA [3], [4] report a zero-shot accuracy between 53% to 97%, and on Waterbirds [5], [6] report a zero-shot accuracy for ViT-B/16 and ViT-B/32 models of around 87.34%. We can split these datasets along some axes and analyze CLIP’s zero-shot prediction.
>
> At the moment, **as an additional experiment in Appendix Section C, we have repeated the similarity analysis from Fig. 3 for CelebA and Waterbirds.** For CelebA, we split along the factors Eyeglasses, Hats as distribution shifts and zero-shot predict the factor Male. For Waterbirds, we can either split based on the background (land vs. water), or based on the combination of bird and land (landbird-on-land/waterbird-on-water vs. landbird-on-water/waterbird-on-land). We observe the same trends (similarity distributions to the train set/LAION differ, similarity is correlated with accuracy) as we did in Sec. 4.
>
> | Dataset | Attributes | Correlation between similarity and zero-shot accuracy $\rho_S$ |
> |-|-|-|
> | CelebA | w/ eyeglasses | 1.0|
> | CelebA | w/ hat | 0.78|
> | | | |
> | Waterbirds | Coregroup | 0.82 |
> | Waterbirds| Land | 0.98 |
>
> Given the rebuttal’s tight time frame and the computational cost of training CLIP, we cannot yet replicate the experiment from Sec. 6 on these datasets. But, if the reviewer feels strongly about additional experiments, we are happy to repeat the complete analysis from Sec. 6 on CelebA and Waterbirds (and possibly also iWILDCam) for the camera-ready version.
>
> ### Question 2: Choice of using the CLIP ViT-B/16 encoder for filtering
>
> **We agree with the reviewer that larger models would potentially capture image features and similarities better**. We originally went with the CLIP ViT-B/16-240+ model as
> 1. it was previously employed by Abbas et al [1] for semantic de-duplication,
> 2. it provides a good balance between performance and computational efficiency [2],
> 3. our nearest neighbor visualizations using this model seemed to be human aligned,
> Having said that, **based on the reviewer’s suggestion, we computed the nearest neighbor similarities of ImageNet-Train to each of the six different datasets using the image embedding space of ViT-L/14 as well — see the newly added Fig. 12 in Appendix Section D**. We then compute the correlation of the ViT-L/14 similarities to ViT-B/16-240+ similarities and observe very high correlation (in the range $\rho=0.8 \dots 0.93$) across datasets (see the newly added Tab. 5 in Appendix Section D). This indicates that the embedding space of ViT-B/16-240+ already sufficiently captures image features and similarities, and that using a bigger model wouldn’t change the trends or our conclusions.
>
> ### Summary
> We trust that our expanded explanation and the supplementary experiments have thoroughly addressed the questions and concerns raised. We sincerely hope this additional information offers a clearer understanding of our work, potentially leading to a reassessment of its evaluation.
>
> [1] https://arxiv.org/abs/2303.09540
>
> [2] https://github.com/mlfoundations/open_clip/blob/91923dfc376afb9d44577a0c9bd0930389349438/docs/openclip_results.csv#L61
>
> [3] https://mmlab.ie.cuhk.edu.hk/projects/CelebA.html
>
> [4] https://openaccess.thecvf.com/content/CVPR2023W/SAIAD/papers/Gannamaneni_Investigating_CLIP_Performance_for_Meta-Data_Generation_in_AD_Datasets_CVPRW_2023_paper.pdf
>
> [5] https://github.com/kohpangwei/group_DRO
>
> [6] https://arxiv.org/pdf/2308.01313.pdf

---

> > ### Comment · Reviewer_fbpP · 2023-11-21
> >
> > Thanks for the authors' response. My rating remains the same. I would encourage authors to test the other proposed datasets despite low zero-shot accuracy. An in-depth analysis of the similarity of images to the LAION dataset might conclude that these datasets are not similar to LAION. Hence, we might need a better training method or dataset so that we can improve zero-shot generalization on these types of datasets as well.

---

> > > ### Author Response · Authors · 2023-11-21
> > >
> > > We highly appreciate your reply and comments. We agree with the reviewer that additional experiments on datasets where CLIP does not perform will strengthen the paper. We will therefore conduct the similarity analysis and pruning experiments on iWILDCam or FMoW for the camera-ready version.

---

### Author Response · Authors · 2023-11-23

Dear reviewers and AC,

We first would like to thank everyone for their time and valuable feedback. We appreciate their assessment of our work as that “****insightful experimental results are presented”.**** Reviewers agree that our experiments are well-defined, insightful, and are of value to the community. In particular, they state that ****“the experiments are clearly defined”**** (fbpP), **“several insightful experimental results are presented”** (MLz9), **“experiments are well motivated and designed, resulting in compelling results**” (fHnd), **“drawn some intriguing conclusions”** (fHnd), and “**similar samples are crucial for the effective robustness of CLIP models”** (X2Xy). Moreover, they also concur that the **“the paper is very well written”** (fbpP), **“the key finding…is helpful”** (MLz9),  **“experimental results are…helpful for the community, especially given that these experiments are very computationally intensive”** (MLz9),  **“… can help with further investigation”** (MLz9).

We received very valuable feedback, incorporated it into the manuscript and uploaded a new version. Moreover, we will conduct additional experiments on iWILDCam or FMoW, and Flickr-30k/MSCOCO for the camera-ready version, as the tight time-frame of the rebuttal didn't allow for retraining CLIP models given our resources. We include a detailed change log for our revised manuscript at the bottom of this post and respond to individual concerns in our individual responses.

### Change log:

- [Reviewer fbpP and fHnd] added additional similarity experimental results and analysis on CelebA and Waterbirds dataset in Appendix C.
- [Reviewer fbpP] added analysis on choice of Similarity metrics in Appendix C.
- [Reviewer MLz9] added an extra discussion section (Section 6)  on the difficulty measuring the true OOD performance of CLIP.
- [Reviewer MLz9] changed “generalization gap” to “similarity gap” throughout the manuscript.
- [Reviewer MLz9] edited abstract and contributions for better clarity.
- [Reviewer fHnd] added more details in Appendix F regarding generating the nearest neighbors visualizations.
- [Reviewer X2Xy] added additional near/ far pruning experimental results on non-ImageNet-like datasets (MNIST/SVHN) in Appendix B2.
- [Reviewer X2Xy] added additional near/far pruning experimental results on ImageNet trained with ResNet in Appendix B3.
- [Reviewers fHnd, MLz9, X2Xy] completely rewrote Sections 4 and 5 to sharpen our core message and improve the flow of the paper; also edited Sections 1 and 6 to improve clarity.
- [Reviewers fHnd, MLz9] added a discussion in Sec. 6 on the role of compositionality for generalization.

---

### Meta-Review · Area_Chair_25wK · 2023-12-08

**Metareview:**

A) The paper studies the strong OOD behavior of CLIP. They hypothesize that clip training sets such as LAION contain examples similar to OOD test sets like imagenet-sketch and imagenet-v2. So the authors remove images similar to these sets from the training set and train a new clip model and they find that while performance does drop a bit, CLIP models still perform well.

B) The paper is experimentally solid and well written.  Both the qualitative examples and quantitive methods used to prune the dataset seem solid. This has been a big question the community has had about its OOD performance and having a large overlap between OOD test sets and training set. The result is interesting! The fact that removing these examples still leads to a model that out-performs imagenet-only training.

C) The paper only uses Imagenet distribution shifts to test OOD performance. The reviewers point out that non imagenet shifts should have been used to further measure clip's OOD performance.

Despite the weaknesses I think the paper's contributions are important and I will vote for acceptance. ImageNet OOD shifts have been very carefully studied [1,2] and are often a good litmus test for model's general OOD robustness, so starting a robustness study there makes a lot of scientific and pragmatic sense.

[1] -Taori, Rohan, et al. "Measuring robustness to natural distribution shifts in image classification." Advances in Neural Information Processing Systems 33 (2020): 18583-18599.

[2]  -  Djolonga, Josip, et al. "On robustness and transferability of convolutional neural networks." Proceedings of the IEEE/CVF Conference on Computer Vision and Pattern Recognition. 2021.

**Justification For Why Not Higher Score:**

The authors could have provided some experiments on non IN test sets to show the findings generalize to other datasets.

**Justification For Why Not Lower Score:**

The paper studies an important problem and is experimentally solid!

---

### Decision · Program_Chairs · 2024-01-16

Accept (poster)